# The hypoxia response pathway promotes PEP carboxykinase and gluconeogenesis in *C. elegans*

Mehul Vora[1], Stephanie M. Pyonteck[1], Tatiana Popovitchenko[1], Tarmie L. Matlack[1], Aparna Prashar[2], Nanci S. Kane[1], John Favate [2], Premal Shah [2] & Christopher Rongo [1,2] ✉

Actively dividing cells, including some cancers, rely on aerobic glycolysis rather than oxidative phosphorylation to generate energy, a phenomenon termed the Warburg effect. Constitutive activation of the Hypoxia Inducible Factor (HIF-1), a transcription factor known for mediating an adaptive response to oxygen deprivation (hypoxia), is a hallmark of the Warburg effect. HIF-1 is thought to promote glycolysis and suppress oxidative phosphorylation. Here, we instead show that HIF-1 can promote gluconeogenesis. Using a multiomics approach, we reveal the genomic, transcriptomic, and metabolomic landscapes regulated by constitutively active HIF-1 in *C. elegans*. We use RNA-seq and ChIP-seq under aerobic conditions to analyze mutants lacking EGL-9, a key negative regulator of HIF-1. We integrate these approaches to identify over two hundred genes directly and functionally upregulated by HIF-1, including the PEP carboxykinase PCK-1, a rate-limiting mediator of gluconeogenesis. This activation of PCK-1 by HIF-1 promotes survival in response to both oxidative and hypoxic stress. Our work identifies functional direct targets of HIF-1 in vivo, comprehensively describing the metabolome induced by HIF-1 activation in an organism.

Hypoxia (oxygen deprivation) plays a central role in diverse human diseases, including ischemic stroke, myocardial infarction, pulmonary hypertension, cerebral palsy, COVID-19, and cancer. In addition to reducing energy (ATP) production by impairing Oxidative Phosphorylation (OXPHOS), hypoxia and subsequent reoxygenation also create oxidative stress by generating toxic Reactive Oxygen Species (ROS). In disorders involving acute hypoxia, both energy deprivation and oxidative stress contribute to tissue damage[1,2].

Metazoans respond to hypoxia using a conserved response pathway. Under aerobic (normoxic) conditions, a prolyl hydroxylase senses oxygen and uses it to covalently modify and negatively regulate hypoxia-inducible factor (HIFα), the transcriptional effector[3] of the pathway (Fig. 1a). When hypoxia ensues, the prolyl hydroxylase is inhibited due to lack of oxygen, resulting in the disinhibition of HIFα.

HIFα becomes stable, dimerizes with HIFβ, and binds to sites throughout the genome to regulate the transcription of specific target genes. Activation of HIF minimizes damage from acute hypoxia; however, long-term adaptations and tissue remodeling triggered by the hypoxia response pathway itself can be detrimental[4,5].

HIF offsets hypoxic damage by regulating multiple physiological processes. HIF upregulates enzymes that mediate glucose uptake and glycolysis, as well as pyruvate dehydrogenase kinase, a negative regulator of metabolite entry into the tricarboxylic acid (TCA) cycle[6]. HIF inhibits OXPHOS by altering subunit composition in the electron transport chain (ETC) and by reducing mitochondrial number[7,8]. The resulting metabolic switch optimizes cells for generating ATP under anaerobic conditions. In certain scenarios (e.g., cancer, dividing stem cells, activated T lymphocytes, endometrial decidualization, and

[1]The Waksman Institute, Rutgers The State University of New Jersey, Piscataway, NJ 08854, USA. [2]The Department of Genetics, Rutgers The State University of New Jersey, Piscataway, NJ 08854, USA. ✉e-mail: crongo@waksman.rutgers.edu

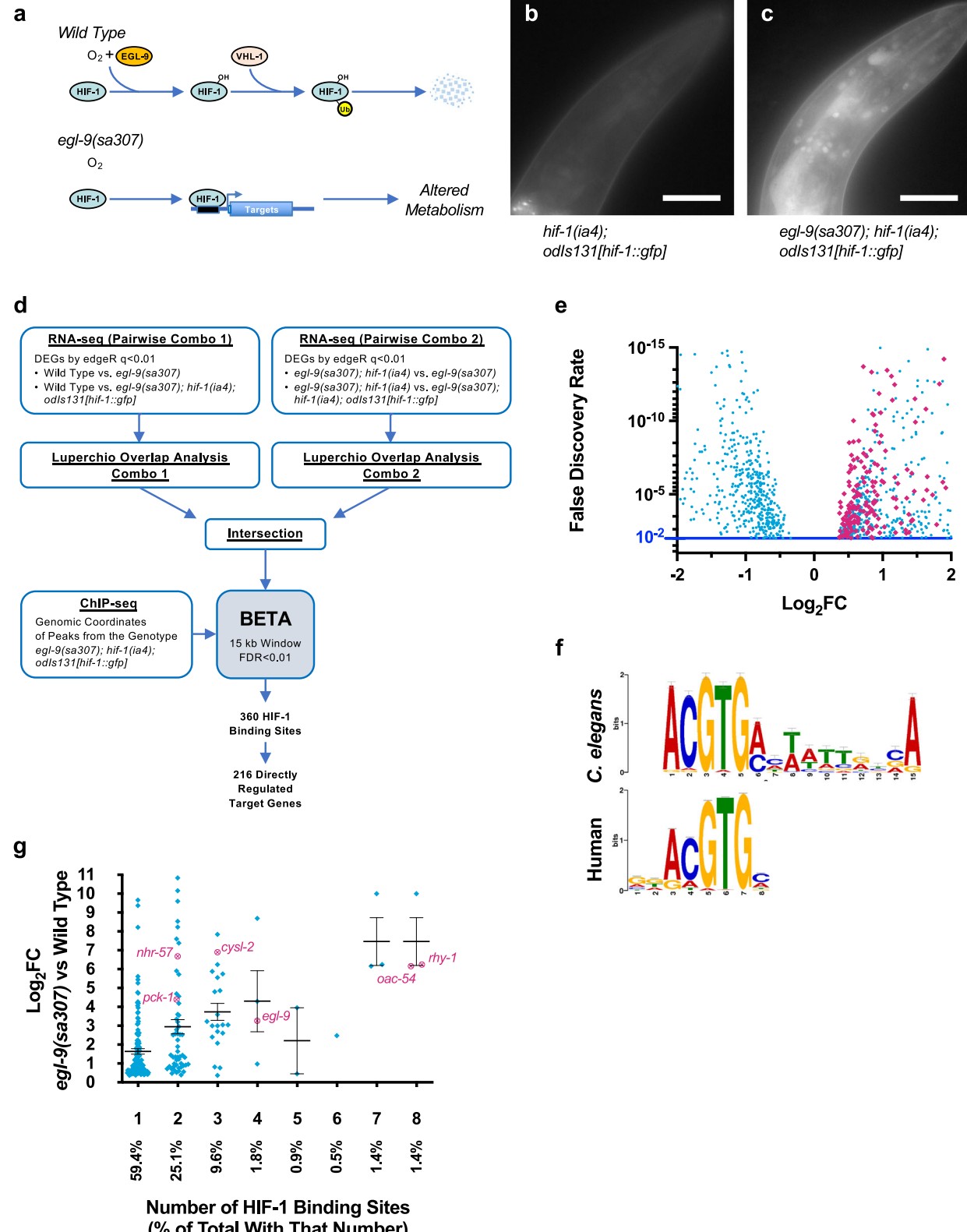

**Fig. 1 | Identifying genes directly regulated by HIF-1. a** Diagram of the hypoxia response pathway in *C. elegans*. HIF-1 is hydroxylated by EGL-9, ubiquitinated by VHL-1, and degraded by the proteasome. In *egl-9(sa307)* mutants, HIF-1 remains stable and regulates the transcription of target genes whose expression alters metabolism. **b**, **c** HIF-1::GFP fluorescence in the indicated genotypes under normoxia. Scale bar indicates 30 microns. Similar results were obtained in three independent trials. **d** Strategy for integrating HIF-1 ChIP-seq and RNA-seq data to identify directly regulated targets. **e** Volcano plot of RNA-seq FDR values versus log₂ fold-change expression for individual genes (blue squares) in *egl-9* mutants relative to wild type. The direct targets identified by BETA are indicated with magenta diamonds. **f** Consensus HRE sequences identified by MEME-suite in humans and enriched in the *C. elegans* ChIP-seq sequences identified by BETA. **g** Graph of log₂ fold change in expression for individual genes in *egl-9* mutants relative to wild type versus the number of HIF-1 binding sites for those genes identified by ChIP-seq. Target genes mentioned in the text are circled and labeled magenta. *N* = 216 total sites. Error bars indicate mean ± SEM.

presumably patients receiving prolyl hydroxylase inhibitors to treat anemia), HIF activates this switch and promotes glycolysis despite aerobic conditions (i.e., the Warburg effect)[6,9–19]. HIF also regulates target genes like EPO and VEGF, which developmentally remodel oxygen delivery[20,21]. We are only just beginning to discover the complete list of HIF targets, determine which of these targets are direct versus indirect, and demonstrate which are relevant in vivo.

Much of our understanding of the hypoxia response has come from studies of cultured cells[22–25]. Here we characterize the adaptive response induced by HIF activation in vivo by using the genetic model system *C. elegans*. These nematodes employ single orthologs of the conserved hypoxia response pathway (the prolyl hydroxylase ortholog EGL-9 and the HIFα ortholog HIF-1), with viable null mutants in *hif-1* and *egl-9* showing altered sensitivity to hypoxic stress[3,26–32]. In addition to regulating survival under hypoxia, the hypoxia response pathway in *C. elegans* also regulates behavior, glutamate receptor trafficking in neurons, mitochondrial dynamics, the egg-laying circuit, and lifespan[27–31,33–38]. HIF-1 is constitutively active in *egl-9(sa307)* mutants, which we use here in an integrated genomic, transcriptomic, and metabolomic approach to obtain a holistic and functional description of HIF activation in an intact animal. Whereas the role of HIF in promoting glycolysis is well appreciated, we find that HIF-1 also promotes gluconeogenesis and the generation of antioxidants through the activation of the PCK-1 PEP carboxykinase. HIF-1, working through PCK-1, promotes survival in response to both oxidative and hypoxic stress. Our work identifies direct functional targets of HIF-1 in vivo, comprehensively describing the metabolome induced by HIF-1 activation in an organism.

## Results

### Identifying direct HIF-1 targets

To identify sites in the genome bound by HIF-1, we created and introduced an *odIs131[hif-1::gfp]* transgene expressing chimeric HIF-1::GFP into *hif-1(ia4)* null mutants and found that it is expressed at levels similar to that of endogenous *hif-1* (Supplementary Fig. 1a–e). Using an *egl-9* null mutant background to activate HIF-1 under aerobic conditions (Fig. 1a), we observed little fluorescence from *hif-1(ia4); odIs131[hif-1::gfp]* animals (Fig. 1b), but clear nuclear HIF-1::GFP fluorescence in essentially all tissues in *egl-9* mutants (Fig. 1c). Moreover, *odIs131[hif-1::gfp]* substituted for endogenous *hif-1* in multiple mutant phenotypes (Supplementary Fig. 1f–i)[28,29,33–35], providing a functional tool to identify genomic binding sites under physiological conditions.

To identify genes directly regulated by HIF-1, we performed ChIP-seq on L4-stage (pre-fertile) *egl-9(sa307); hif-1(ia4); odIs131[hif-1::gfp]* animals using anti-GFP antibodies and high-throughput sequencing. We identified 604 sequencing read peaks corresponding to HIF-1::GFP binding sites (Supplementary Data 1), most of which fell within 500 base pairs (bps) of a nearby gene and were enriched for the human hypoxia response element (HRE, Supplementary Fig. 2a–c)[21,39].

The nearest gene to a ChIP-seq peak is not always the direct target regulated by the associated transcription factor binding site. We therefore performed RNA-seq to analyze differential gene expression due to HIF-1 activation, using these data to identify HIF-1-regulated genes near HIF-1 binding sites. We examined transcriptomes of L4-stage animals in four distinct pairwise combinations of genotypes in which HIF-1 is active versus inactive, considering two experimental comparisons that would be enriched for HIF-1-regulated differentially expressed genes (DEGs): *egl-9* mutants (active HIF-1) vs. wild-type (inactive HIF-1), and *egl-9* mutants (active HIF-1) vs. *egl-9 hif-1* double mutants (inactive HIF-1) (Fig. 1d and Supplementary Fig. 3a, b). For each of these two experimental comparisons, we took advantage of the *odIs131[hif-1::gfp]* transgene, which rescues *hif-1* mutants, to provide an additional experimental dataset that reduced unrelated genetic background effects. Our pairwise approach ensures that the intersection of the comparisons contain high-stringency

HIF-1-regulated genes (Supplementary Fig. 3c, d and Supplementary Data 2).

One shortcoming of determining DEGs from the intersection (overlap) of separate analyses is that the reliance on the arbitrary threshold for each analysis can lead to an underestimation of the size of the overlap. We therefore used Luperchio Overlap Analysis (LOA)[40] to identify DEGs, using evidence of potential DEGs in the pairwise comparison lacking the *odIs131* transgene to inform the state of those potential DEGs in the pairwise comparison containing the transgene (Fig. 1d). We reasoned that the highest confidence HIF-1-regulated DEGs would be enriched in the intersection of *egl-9* mutants compared to wild type (LOA Combo 1) and in *egl-9* mutants compared to *egl-9 hif-1* double mutants (LOA Combo 2). We analyzed the resulting list of DEGs with BETA software[41], which infers direct target genes by integrating DEGs and ChIP-seq data, to identify 216 differentially expressed direct target genes for 360 nearby HIF-1 ChIP-seq binding sites (Fig. 1d and Supplementary Data 3). Every identified direct target was exclusively upregulated when HIF-1 was active, consistent with a transcriptional activator (Fig. 1e). Direct targets were enriched for HRE sequences (Fig. 1f). We identified genes whose expression was previously shown to be regulated by HIF-1[28,29,33,34], further validating our approach (Fig. 1g). More than half (59%) of the HIF-1 targets were associated with a single binding site, and the magnitude of expression loosely correlated with their number of binding sites (Fig. 1g). Two key negative regulators of HIF-1 (*rhy-1* and *egl-9*)[28] contained some of the highest numbers of HIF-1 binding sites and were dramatically upregulated by HIF-1, indicating a strong negative feedback loop as part of the response (Supplementary Fig. 3e).

### Characterizing HIF-1 binding sites

To determine if the HIF-1 binding sites for these targets were functional, we generated fluorescent Venus transcriptional reporter transgenes (Fig. 2a and Supplementary Fig. 4a) for one novel target, *pck-1* (PEP carboxykinase) and one established target, *rhy-1*. Elevated levels of Venus fluorescence were observed from the transgenes in *egl-9* mutants, in which HIF-1 is active, compared to wild type and *egl-9; hif-1* double mutants, in which HIF-1 is inactive (Fig. 2b–g and Supplementary Fig. 4b). Transgenes in which either the complete sequence encoding the HIF-1 ChIP-seq peak was removed (ΔPeak), the 6-bp core HRE in the peak was deleted (ΔHRE), or only a minimal promoter sequence was present gave reduced reporter expression compared to that of the wild-type promoter (Fig. 2h and Supplementary Fig. 4b), showing that HREs are required in vivo for HIF-1 to regulate target gene expression.

The ModENCODE/ModERN consortium previously identified High Occupancy Target (HOT) sites where occupancy by multiple (>15) transcription factors occurs; this phenomenon is possibly due to HOT sites being "sticky" regions of the genome that lead to false signals within ChIP-seq data[42,43]. We surveyed the known binding sites of all transcription factors within the ModENCODE/ModERN database for overlap with HIF-1 sites. An examination of all 604 HIF-1 binding sites revealed most to be in or near known HOT sites (Supplementary Fig. 4c), with little differential regulation observed in nearest neighbor genes (Supplementary Fig. 5a). By contrast, only about half of the functional HIF-1 sites identified using LOA and BETA were at HOT sites (Supplementary Fig. 4c), and those genes associated with the remaining HOT sites showed HIF-1-dependent changes in gene expression, suggesting they are functional (Supplementary Fig. 5b–e). Violin histograms (Fig. 2i) showed that the use of LOA, BETA, and stringent RNA-seq data to identify targets regulated by HIF-1 sites resulted in an increase in target calling accuracy, with a shift from HOT sites to low occupancy sites (<5 other factors bound), even with just the use of BETA alone. AHA-1 binding sites were enriched near HIF-1 sites (Fig. 2j), as expected for a HIF-1 dimerization partner[3]. Sites bound by SKN-1, the *C. elegans* Nrf2 ortholog that promotes an antioxidant

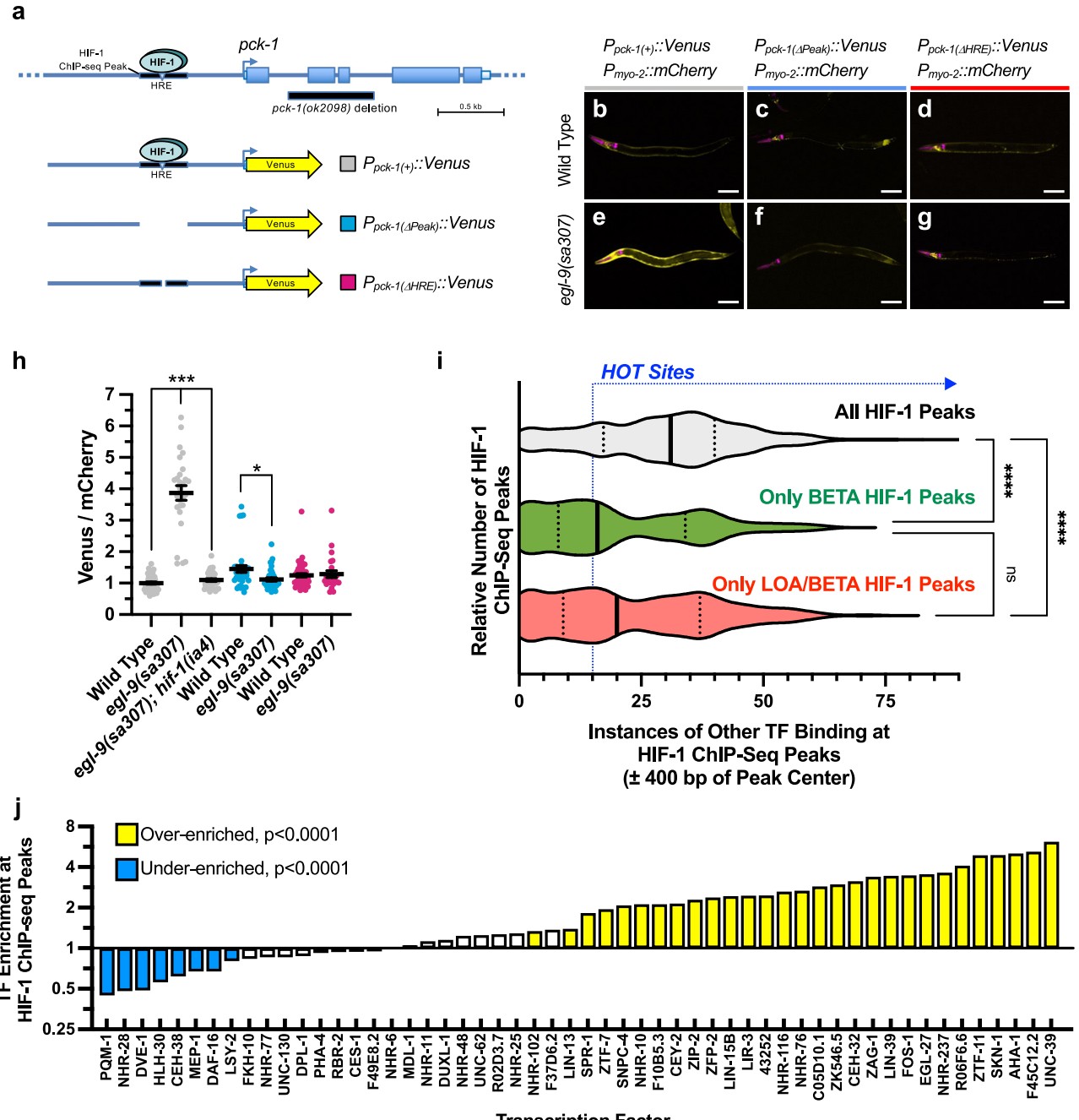

**Fig. 2 | Characterizing HIF-1 binding sites. a** Diagram of the *pck-1* locus and the different promoters used to drive Venus expression. The oval indicates HIF-1, and the black line beneath it indicates the HIF-1 binding site identified by ChIP-seq. The inverted triangle indicates the site of the HRE motif. Arrows indicate the TSS. Boxes indicate exons. The yellow arrow indicates sequences encoding the fluorescent Venus reporter. The black bar under the gene indicates sequences removed in the *pck-1(ok2098)* deletion. **b**–**g** Fluorescence images of animals expressing the transgenes indicated above the images and with the indicated genotype (wild-type, top row; *egl-9* mutant, bottom row) under normoxic conditions. Venus fluorescence is indicated in yellow. Fluorescence from mCherry, used as an internal control in which fluorescence is not expected to change as it is not regulated by HIF-1, is magenta. Scale bar indicates 100 microns. Similar results were obtained in three independent trials. **h** Graph of Venus/mCherry fluorescence ratios for the indicated genotype under normoxic conditions. Dot color indicates a specific reporter as per panel (**a**) Error bars indicate mean ± SEM. \*\*\**P* < 0.001, \**P* < 0.05 ANOVA/Bonferroni Multiple Comparison two-sided test for the indicated comparisons. *N* = 267 total

animals (25–51/column). **i** Violin plot of histogram counting the number of HIF-1-binding sites (ChIP-seq peaks) against the number of other transcription factors (TFs) known to bind to each site's region of the genome (within 400 bps). The top violin plot indicates all HIF-1 binding sites, whereas the middle (green) and bottom (red) violin plots indicate only the functional HIF-1 binding sites identified by BETA alone or the combined LOA and BETA approaches, respectively. The solid line indicates median, whereas dotted lines indicate quartiles. HOT sites (peaks near the binding site of 15 or more other transcription factors) are indicated by the blue dotted line. \*\*\*\**P* < 0.0001, \**P* < 0.05 ANOVA/Kruskal–Wallis Multiple Comparison two-sided test for the indicated comparisons. **j** Graph of the fold change in enrichment (log2) of each indicated transcription factor bound within 400 bps of a HIF-1 binding site (ChIP-seq peak center). TFs with overrepresented or under-represented binding near HIF-1 sites compared to chance (*P* < 0.0001 via a one-sided Bootstrap test adjusted for multiple comparisons via Benjamini–Hochberg using FDR values based on $1 \times 10^5$ bootstrapped datasets) are highlighted in yellow or cyan, respectively. TF data are from the modERN/ModENCODE consortium.

**Table 1 | Gene ontology of 216 targets directly upregulated by HIF-1**

| Process | Number of genes | Fold enrichment | FDR |
|---|---|---|---|
| **Glucose metabolism** | 11 | 30.6 | $5.6 \times 10^{-10}$ |
| Glycolysis | 7 | 31.8 | $2.8 \times 10^{-06}$ |
| Gluconeogenesis | 10 | 41.7 | $9.9 \times 10^{-10}$ |
| **Amino acid metabolism** | 10 | 5.1 | $5.6 \times 10^{-03}$ |
| Serine biosynthesis | 3 | 50.0 | $8.3 \times 10^{-03}$ |
| Sulfur/cysteine metabolism | 4 | 23.5 | $6.3 \times 10^{-03}$ |
| **Fatty acid metabolism** | 6 | 7.3 | $2.2 \times 10^{-02}$ |
| Beta-oxidation | 4 | 15.4 | $2.4 \times 10^{-02}$ |
| **Response to O$_2$ levels** | 5 | 14.7 | $5.0 \times 10^{-03}$ |
| **Oxidation reduction** | 20 | 3.9 | $3.2 \times 10^{-05}$ |

Enriched Gene Ontology (GO) classifications for the 216 direct HIF-1 targets. Fold enrichment and FDR for each classification and subclassification is indicated.

response[44], were also enriched, suggesting shared targets between the hypoxia and antioxidant responses. By contrast, sites bound by PQM-1, a zinc finger transcriptional antagonist of DAF-16/FOXO, were underrepresented near HIF-1 sites, consistent with the increased hypoxia survival observed in *pqm-1* mutants[45]. Finally, the use of LOA and BETA allowed the identification of regulatory sites acting distal from the TSS of regulated target genes (Supplementary Fig. 2a, b).

## HIF-1 reprograms metabolism

For the 216 direct targets upregulated by HIF-1, GO terms for glycolysis, gluconeogenesis, amino acid metabolism, sulfur oxidation, fatty acid beta-oxidation, and oxidation-reduction metabolism showed enrichment (Table 1), suggesting that HIF-1 reprograms metabolism by directly binding to the promoters of key metabolic pathway enzymes. We obtained metabolomic profiles to determine if the transcriptionally upregulated pathways were more populated with metabolites from those pathways when HIF-1 was active under aerobic conditions compared to when it was not active. We observed differences ($P < 0.05$) in the levels for 175 metabolites, with HIF-1 activation resulting in elevated levels of various amino acids, carbohydrates, lipids, and nucleotides, suggesting changes in multiple metabolic pathways (Fig. 3a and Supplementary Data 4).

To examine metabolite levels in the context of specific biochemical pathways, we mapped wild-type versus *egl-9* mutant metabolome data against corresponding RNA-seq data for enzymes that catalyze reactions involving each specific metabolite. As expected[46,47], HIF-1 directly promoted the expression of multiple key glycolysis enzymes and increased the metabolite population of this pathway (Fig. 3b). Most TCA cycle enzymes and metabolites were unchanged in *egl-9* mutants (Fig. 3c). However, *egl-9* mutants had elevated levels of *mdh-1* (malate dehydrogenase) and *ldh-1* (lactate dehydrogenase), as well as their associated metabolites malate, pyruvate, and lactate (Fig. 3d). Elevated malate and malate dehydrogenase are indicative of activity in the glyoxylate cycle, a variation of the TCA cycle used in some organisms to convert citrate to oxaloacetate to feed the gluconeogenesis pathway.

We also observed indicators of increased gluconeogenesis (Fig. 3b). As previously discussed, HIF-1 directly and dramatically promotes the expression of *pck-1* (PEP carboxykinase), which catalyzes a key rate-limiting step specific to gluconeogenesis in which oxaloacetate (OA) is converted to phosphoenolpyruvate (PEP). In addition to regenerating glucose, gluconeogenesis provides substrates for the pentose phosphate pathway (PPP), which generates reducing equivalent NADPH, 5-carbon sugars (e.g., ribose-5-phosphate used to synthesize nucleic acids), and erythrose-4-phosphate (used to synthesize

aromatic amino acids). We observed an increase in PPP metabolites in *egl-9* mutants relative to wild type (Fig. 3e), although none of the key enzymes within the PPP were direct targets of HIF-1, suggesting that increased PPP flux is an indirect effect of HIF-1 activation of gluconeogenesis.

NADPH generated from the PPP serves as a reducing equivalent for fatty acid synthesis, and both synthesized and dietary fatty acids are attached to glycerol-3-phosphate (G3P) for conversion to storage lipids. Consistent with this metabolic pathway, we observed decreased G3P and increased levels of certain fatty acids when HIF-1 is active (Fig. 3f). Phospholipids are also synthesized from G3P, and we found that active HIF-1 increased the levels of certain phospholipid species (Fig. 3f).

NADPH also replenishes reduced glutathione, which is a major antioxidant for combating oxidative stress. We observed higher levels of glutathione in *egl-9* mutants relative to wild-type (Fig. 3g). Indeed, many of the direct target genes most dramatically upregulated by HIF-1 lie within pathways that produce glutathione (Fig. 3g), and the catalases and superoxide dismutases that combat oxidative stress are all indirectly upregulated (Fig. 3h). Oxidative stress response pathways also help fight bacterial infection in *C. elegans*, as bacterial pathogens like *Pseudomonas aeruginosa* produce toxins like HCN and H$_2$S[47,48], which can be neutralized by these pathways. Indeed, we observed that HIF-1 directly promotes the expression of the entire H$_2$S/HCN detoxification pathway (Fig. 3i), including *cysl-2* (cyanoalanine synthase) and *sqrd-1* (sulfide quinone oxidoreductase), which converts these toxins to polysulfides and sulfates[47]. Upregulation of H$_2$S/HCN detoxification by HIF-1 is consistent with *egl-9* and *hif-1* mutants being resistant and sensitive, respectively, to *Pseudomonas* infection and sulfide/cyanide toxicity.

## PCK-1 promotes survival during hypoxic stress

Traditionally, HIF-1 is thought to protect against hypoxia by promoting anaerobic ATP synthesis. Our multiomics analyses also highlight its promotion of gluconeogenesis and the oxidative stress response in *egl-9* mutants. To test whether direct HIF-1 targets are also upregulated by hypoxia, we used qRT-PCR to measure mRNA levels in either wild-type or *hif-1* mutants (Fig. 4a and Supplementary Fig. 6). All tested target genes were upregulated by hypoxia in a HIF-1-dependent manner, including *pck-1*, the rate-limiting enzyme of gluconeogenesis.

Given that both hypoxia and subsequent reoxygenation result in the production of reactive oxygen species (ROS) and oxidative stress, we reasoned that the upregulation of PCK-1 by HIF-1 might also protect against hypoxia by mobilizing an antioxidant response through gluconeogenesis and the PPP. To rule out metabolic contributions from the live *E. coli* used to grow nematodes, we fixed bacterial cultures with formaldehyde to render them metabolically inert[49]. We grew nematodes on fixed *E. coli* until they reached the L4 stage, then exposed these animals to 48 h of hypoxia without food, measuring survival after 24 h of recovery at normoxia. Mutants for *pck-1*, like *hif-1* mutants, were viable under normoxic conditions (Supplementary Fig. 7a). However, both *hif-1* and *pck-1* single mutants were significantly susceptible to hypoxia compared to wild-type (Fig. 4b), and this susceptibility was rescued by supplementation with PEP (the product of the PCK-1-catalyzed reaction that is elevated when HIF-1 is active, as per Fig. 3b) but not the upstream metabolite pyruvate. We observed comparable results with a hypoxia survival assay for embryo hatching (Fig. 4c and Supplementary Fig. 7b). The antioxidants glycolate and N-acetyl cysteine (NAC) can reduce oxidative stress when fed to *C. elegans*[50,51], and we found that supplementation with these antioxidants rescued *hif-1* and *pck-1* mutants for survival during hypoxia (Fig. 4d, e and Supplementary Fig. 7c). Similar results were observed with nematodes grown on live *E. coli* (Supplementary Fig. 7d, e). In addition, we found that *pck-1* mutants are sensitive to oxidative stress,

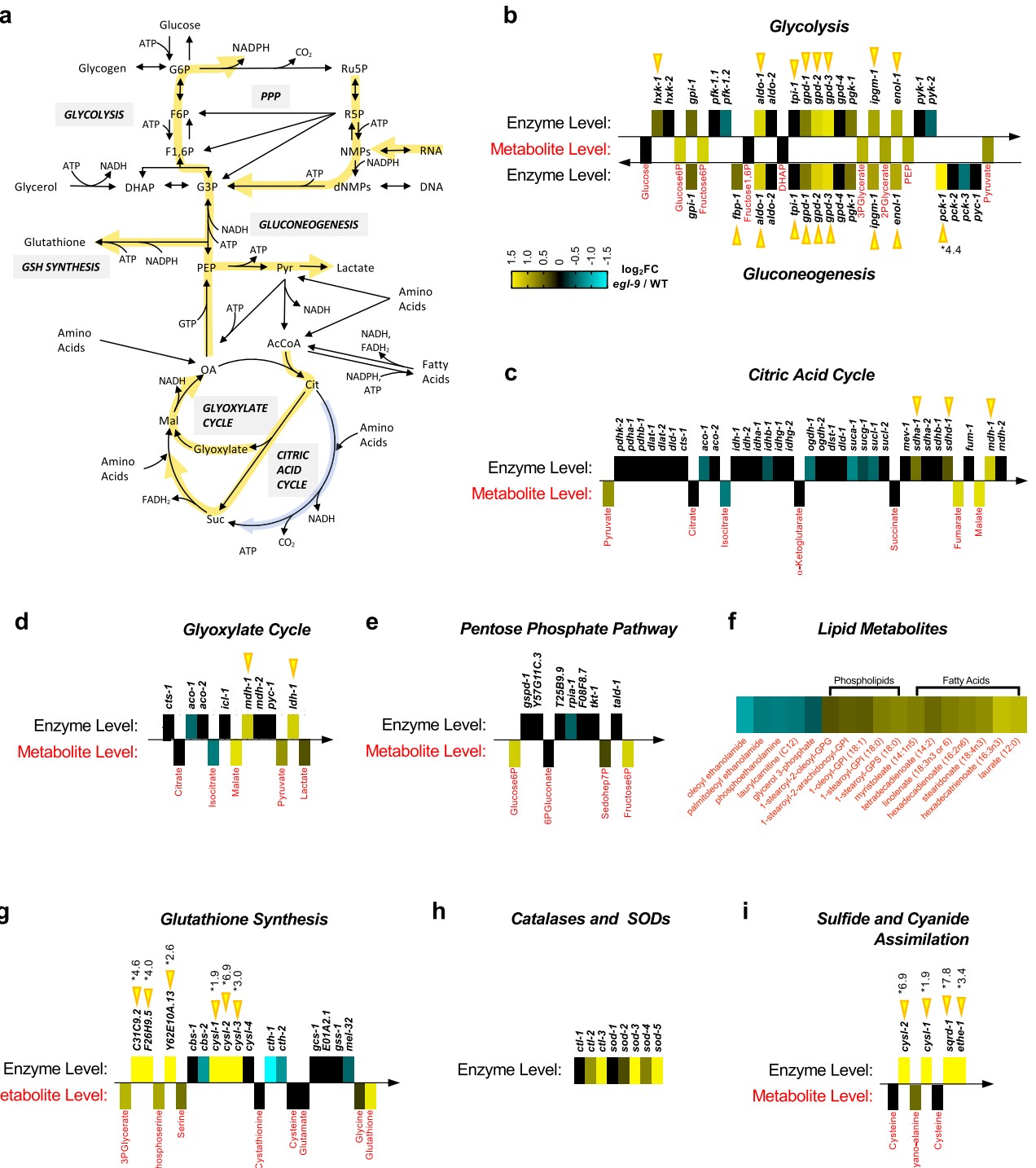

**Fig. 3 | HIF-1 reprograms metabolism. a** Overview of major metabolic pathways impacted by HIF-1 activation. Yellow and cyan arrows overlaying the pathways indicate upregulation or downregulation, respectively, in *egl-9* mutants relative to wild type. **b**–**i** Combined heatmap for metabolites (red text) and the enzymes (black text) and directional pathways (arrows) known to regulate their metabolism. The color index in (**b**) is applicable to all panels in the figure. On one side of each pathway arrow (above for most arrows except gluconeogenesis), differentially colored boxes indicate enzyme mRNA expression levels (log₂ fold change) in *egl-9* mutants relative to wild type; enzymes are labeled in the black text by their indicated *C. elegans* gene name. On the other side of each pathway arrow (below for most arrows except gluconeogenesis), differentially colored boxes indicate metabolites levels (log₂ fold change) in *egl-9* mutants relative to wild-type; metabolites are labeled in red text. Gene expression changes that exceed a log₂ fold change of 1.5 are marked with an asterisk showing the actual log₂ fold-change value. Yellow arrowheads indicate genes with HIF-1 binding sites and that show direct regulation by HIF-1. Glycolysis and gluconeogenesis are graphed together, in opposite direction, as they share multiple metabolites and enzymes.

as they showed poor survival to exposure to the superoxide generator paraquat (Supplementary Fig. 7f). Mutations in *pck-1* did not suppress the egg retention or extended lifespan phenotypes of *egl-9* mutants (Supplementary Fig. 8), indicating that the upregulation of PCK-1 does not mediate all functions of HIF-1. Taken together, our results demonstrate that the direct promotion of gluconeogenesis and oxidative stress resistance by HIF-1 helps animals survive hypoxia.

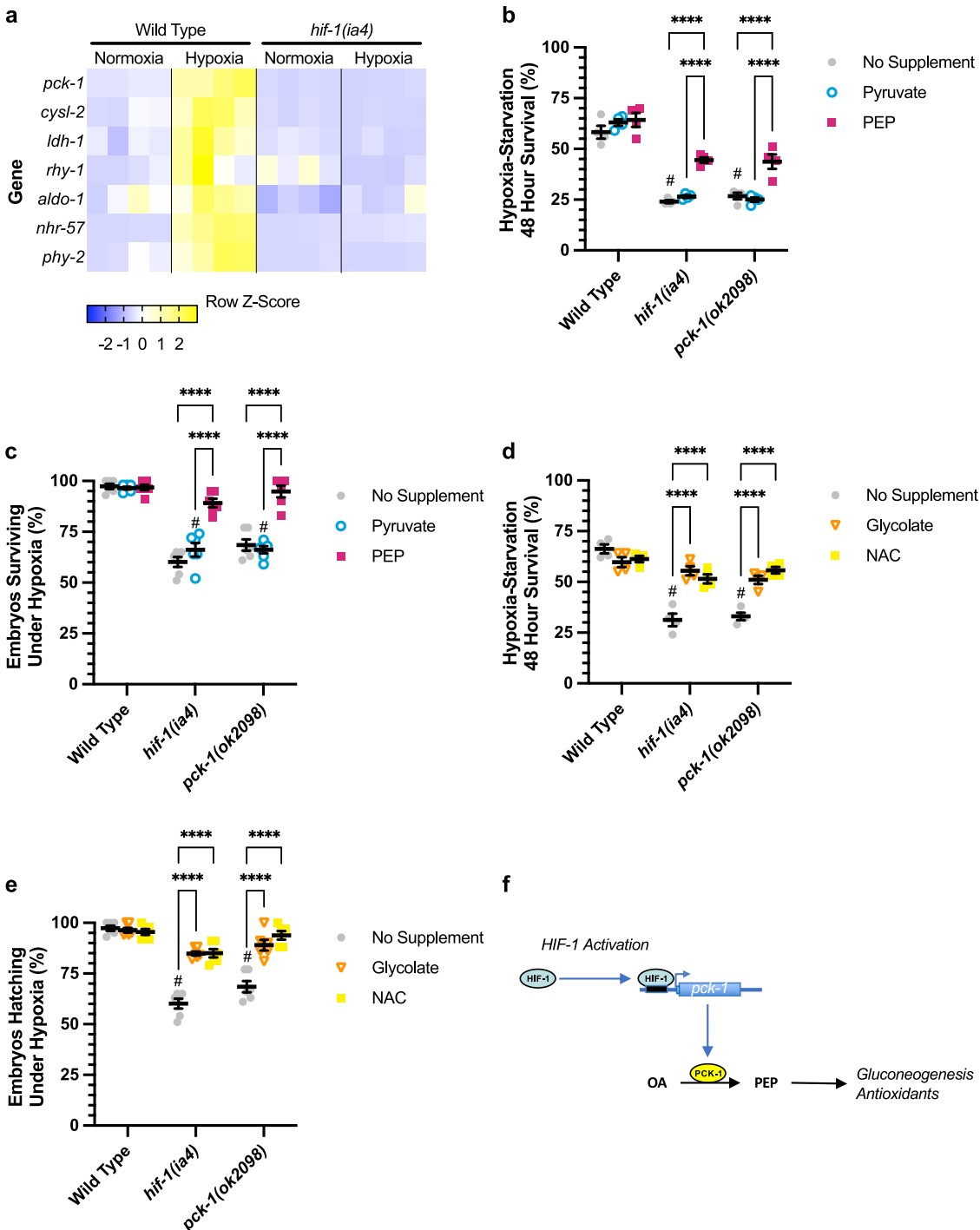

**Fig. 4 | HIF-1 targets required for adaptive survival. a** Heatmap of row Z-scores for qRT-PCR measurements from L4-stage animals of transcript levels for the indicated genes (rows) as normalized to actin. Columns indicate the genotypes (either wild type or *hif-1(ia4)* mutants) and experimental conditions (4 h under either normoxia or hypoxia (0.5% O₂), with four replicates from each genotype/condition shown. **b, d** Percent of L4-stage animals, grown on fixed *E. coli*, to survive liquid culture without food and under hypoxia (0.5% O₂, 48 h at 25 °C, with 24 h recovery at 20 °C). Error bars indicate mean ± SEM. **c, e** Percent of embryos, obtained from animals grown on fixed *E. coli*, to survive 24 h of hypoxia at 25 °C, with 24 h recovery at 20 °C. Animals are indicated by colored symbols, and associated figure legends list the condition in which they were grown: on plates supplemented with either metabolites or antioxidants. Error bars indicate mean ± SEM. For graphs (**b–e**), ****$P < 0.0001$, ***$P < 0.001$, **$P < 0.01$, *$P < 0.05$ ANOVA/Sidak's multiple comparison two-sided test between each supplement and the no-supplement control. #$P < 0.0001$ ANOVA/Sidak's multiple comparison two-sided test between the indicated mutant versus wild-type, both no-supplement controls. Data for each column in (**b, c**) represents four biological replicates, and in (**d, e**) represents six biological replicates, typically 20–100 animals per replicate. **f** Model illustrating how HIF-1 binds near the *pck-1* promoter to upregulate PCK-1 expression and drive gluconeogenesis and the production of antioxidants.

## HIF-1 direct targets are conserved

We identified putative human sequence orthologs for the direct targets identified in our *C. elegans* analysis (Supplementary Data 5). We reanalyzed published RNA-seq data from human cells undergoing HIF1A activation and identified several clusters of orthologs showing correlated patterns of co-expression (Supplementary Fig. 9). Although only 0.3% of human genes showed consistent upregulation by HIF1A, 6.8% of putative human orthologs to

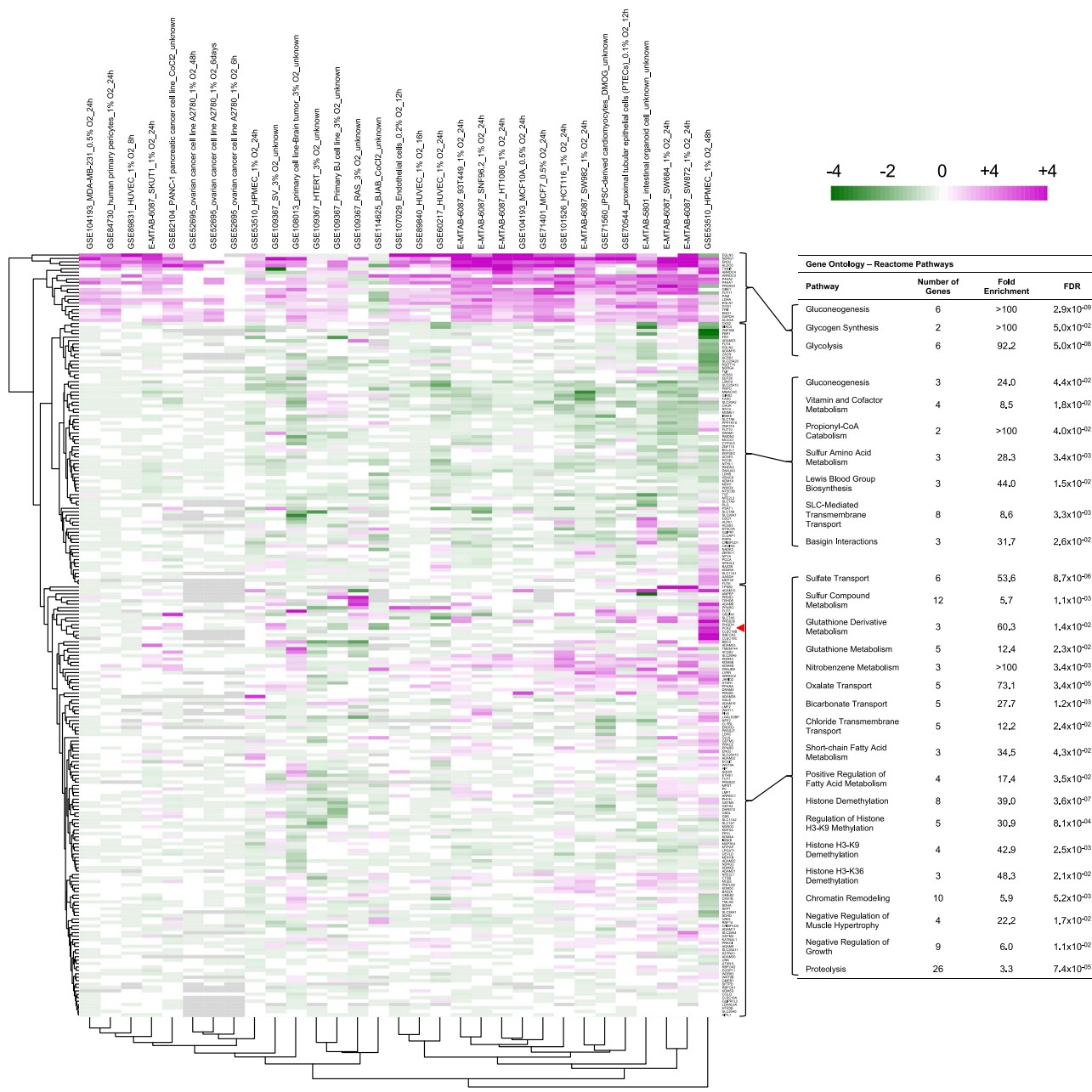

**Fig. 5 | Differential gene expression for human orthologs of HIF-1 direct targets across multiple independent experiments.** Heatmap of mRNA expression across 31 independent experiments (columns, including accession numbers and hypoxia treatment indicated in vertical names) involving HIF1A activation. Fold change in expression (expressed as log2) is mapped for individual human orthologs (rows) of the *C. elegans* HIF-1 direct targets identified in this study. Only genes showing expression in at least 15 experiments are shown. Genes with similar patterns of expression across experiments are hierarchically clustered. Clusters of genes were analyzed for GO Reactome Pathway enrichment. While a cluster of genes (top of map) that were enriched for glycolysis and gluconeogenesis GO and reactome annotations showed near-universal upregulation across all the experiments, most putative orthologs clustered based on shared context- or tissue-dependent expression, including PCK2 (red arrowhead), an ortholog of *pck-1*. These latter clusters were enriched for sulfur and glutathione metabolism, metabolite transport, and chromatin modification.

*C. elegans* HIF-1 direct targets showed consistent upregulation, a 23-fold enrichment (*P* value < $1.0 \times 10^{-16}$, proportion test). A cluster of orthologous genes were upregulated by HIF1A across nearly all studies, with GO enrichment similar to that observed for their nematode counterparts (Fig. 5). Other clusters showed upregulation in subsets of experiments. Indeed, PCK2, an ortholog of *C. elegans pck-1*, belonged to such a cluster that also included genes involved in cysteine and glutathione metabolism, suggesting that this PEP carboxykinase is a key HIF1A regulatory target depending on tissue or context.

## Discussion

Stress response pathways coordinate the regulated expression of multiple target genes needed to rebalance physiological homeostasis. To meet the energy needs of an organism in a hypoxic environment, HIF promotes ATP synthesis by promoting glycolysis. However, the nature and role of the full array of HIF targets in vivo are not understood. Our studies combined ChIP-seq and RNA-seq to identify the full array of direct HIF-1 targets in *C. elegans*, combined with metabolomics to understand the physiological changes resulting from their expression. HIF-1 reprograms metabolism beyond simply augmenting

glycolysis as previously thought. We show here that it directly promotes lipid metabolism, the glyoxylate cycle, gluconeogenesis, the PPP, $H_2S$/HCN detoxification, and glutathione synthesis. This study combines genomic, transcriptomic, and metabolomic approaches to characterize the action of HIF activation in vivo.

A simple approach to discern true functional targets of transcription factors (TFs) like HIF-1 has been to integrate binding site identification, typically from ChIP-seq experiments, with transcriptomics analysis, typically from RNA-seq analysis comparing conditions in which a given transcription factor is active versus inactive. Lists of nearest neighbor genes from ChIP-seq analysis are compared to lists of differentially expressed genes (DEGs) from RNA-seq analysis, with the intersection of those lists deemed functional targets. However, assigning a gene as a target of a transcription factor based solely on ChIP-seq binding data and TSS proximity has limitations. This "nearest neighbor" approach does not assess changes in expression in response to TF activity for each putative target. This problem is compounded in organisms with compact genomes like those of *C. elegans* or *Drosophila*, where multiple genes are often found adjacent to a given ChIP-seq peak. Thus, any targets identified by this approach might not be true functional targets.

A more powerful approach is to use software like the BETA algorithm, which uses the DEG patterns derived from RNA-seq studies that compare mRNA levels when a TF is active versus inactive, to determine which gene near the binding site of the given TF (identified from ChIP-seq) is the target[41]. BETA prioritizes functional expression differences in addition to proximity, increasing its ability to identify true functional targets. Although stringent, this approach can lead to an underestimation of the size of the overlap since an arbitrary significance threshold or rank must be assigned to each of the separate analyses. To overcome this drawback, we combined BETA with an overlap analysis methodology (Luperchio Overlap Analysis or LOA[40]) that allows multiple comparisons to provide evidence to each other regarding differential gene expression to overcome this limitation. The pretext of this analysis is that a set of DEGs in a given condition (e.g., wild type versus the *egl-9* mutant) is informative about the state of the same genes in another similar condition (e.g., wild-type versus the *egl-9* mutant that also contains a *hif-1* mutation and the rescuing *hif-1::gfp* transgene). By utilizing this approach, we were able to enlarge the number of functional targets compared to using the more stringent intersection approach. Our combined methodology allowed for the identification of HIF-1 binding sites that were more distal to the TSS of the regulated gene (Supplementary Fig. 2a, b).

Another complication in identifying functional TF binding sites is the presence of high-occupancy target (HOT) sites, which are over-represented binding peaks found across multiple independent ChIP-seq studies, scattered across the genome[43]. Most HOT sites are either ChIP-seq artifacts or binding sites with no clear functional role, at least with respect to regulating the transcription of neighboring genes. Here again, our combined BETA and LOA approach overcame this limitation. We showed that the target genes identified by BETA and LOA are more specific, as the number of targets attributed to HOT sites within the genome is reduced compared to both BETA alone (intersection) and nearest neighbor approaches (Fig. 2i). In addition, BETA and LOA were able to distinguish functional HOT sites (i.e., sites with nearby genes that show HIF-1-dependent differential regulation) from non-functional HOT sites (Supplementary Fig. 5). Genes near non-functional HOT sites, as identified by the nearest neighbor approach, did not show any GO term enrichment (PANTHER analysis, FDR < 0.05). By contrast, genes near functional HOT sites, as identified by BETA and LOA, showed GO term enrichment similar to that of genes near non-HOT sites, including for glycolysis, gluconeogenesis, and amino acid metabolism, further demonstrating the power and utility of this approach. For researchers designing new experiments to identify the direct functional targets of a transcription factor, we recommend that the experimental design contain at least two comparative conditions for DEG identification, with subsequent analysis of the DEG lists from those comparisons performed by a combination of LOA and BETA.

The HIF-1 target genes we identified were enriched with factors involved in metabolism, which we attempted to validate through metabolomics. Metabolomic analysis only provides a snapshot in time of the metabolic state in each sample. True metabolic flux analysis requires experiments that follow labeled metabolites, which cannot be done in *C. elegans*. Changes in metabolic flux can be modeled, but such models require assumptions that limit their utility. Nevertheless, metabolomic analysis such as the kind performed in this work creates testable hypotheses that can be explored through experimentation. We observed that HIF-1 directly promotes the expression of the *pck-1* PEP carboxykinase, a key mediator of gluconeogenesis, as well as metabolic flux through the pathway (Fig. 4f). Gluconeogenesis provides the metabolites needed to make glutathione and produce NADPH reducing equivalents required to combat ROS and oxidative stress. Indeed, we found that *pck-1* mutants survive hypoxic stress as poorly as *hif-1* mutants, and both mutants are rescued by supplemental PEP, the product of PCK-1. In addition, supplemental antioxidants rescue hypoxia survival of these mutants, and HIF-1 activation can offset damage by agents that cause oxidative stress. Our meta-analysis of human expression profile data suggests that PCK2, an ortholog of *pck-1*, is part of a group of new, context-dependent targets regulated by HIF1A (Fig. 5). Our results highlight the need for HIF-1 to not only promote anaerobic energy production, but also to mobilize antioxidant defenses through gluconeogenesis. Indeed, blocking gluconeogenesis specifically might present an approach to treating HIF-1-positive tumors.

Although a connection between hypoxia and gluconeogenesis has been made before[52], our findings go beyond simply showing that HIF-1 directly binds to and regulates the expression of a key mediator of gluconeogenesis. Instead, our work highlights the role of the gluconeogenesis-PPP metabolic chain in offsetting oxidative stress during hypoxia, and it shows that restoring redox homeostasis is an underappreciated physiological outcome of the HIF-1/gluconeogenesis regulatory interaction important for the survival of hypoxic stress. This finding has implications not only for diseases involving hypoxia (e.g., cancer, heart disease, stroke, COPD, cerebral palsy, pulmonary hypertension, COVID-19, et cetera), but for our understanding of the evolution of aerobic organisms and their ability to adapt to terrestrial and aquatic habitats with variable oxygen availability.

## Methods

### Generation of transgenes and transgenic animals
*C. elegans* strains were derived from the N2 strain, with hermaphrodites analyzed in all experiments. Strains were obtained from the *C. elegans* Genetics Center. Nematodes were cultured on OP50 *E. coli* seeded on NGM plates unless otherwise stated. The GFP-tagged *hif-1* transgene was obtained as a genomic fosmid from the TransgeneOme Project[53]. The polyclonal stab culture was streaked out under triple selection (chloramphenicol, streptomycin, nourseothricin), and individual clones were selected for transgene construct validation by sequencing. The construct was introduced into the germline genome of *hif-1(ia4)* null mutants and then stably integrated into the genome using microparticle bombardment[54]. The resulting stably integrated line *odIs131* was made homozygous by selecting single hermaphrodites that gave 100% Non-Unc progeny, then outcrossed four times. Subsequently, the *unc-119(ed3); odIs131* strain was crossed to *unc-119(ed3); egl-9(sa307); hif-1(ia4)* to generate the following strains used for ChIP-seq: OR3349 *unc-119(ed3); hif-1(ia4); odIs131[hif-1::gfp, unc-119(+)]* and OR3350 *unc-119(ed3); egl-9(sa307); hif-1(ia4); odIs131[hif-1::gfp, unc-119(+)]*.

Fluorescent Venus transcriptional reporter transgenes containing genomic sequences encoding the HIF-1 binding site and the transcriptional start site were generated to validate some of the direct targets. To quantify expression, a $P_{myo-2}$::mCherry transgene was introduced into the genome in conjunction with the pck-1::Venus or rhy-1::Venus transgenes. As myo-2 expression did not vary in the RNA-seq datasets, Venus expression was normalized to mCherry expression in the pharynx. At least two independent lines were examined for each experiment.

To generate $P_{pck-1}$::Venus transgenic animals, 2816 bp upstream of the ATG of pck-1 was amplified and cloned upstream of a Venus coding sequence present in pPD95.77-mVenus. Using Q5-site directed mutagenesis (Life Technologies Ltd), the ΔPeak (1084 – 912 upstream of start) and the ΔHRE (1019 – 1003 upstream of start) were deleted to yield $P_{pck-1(ΔPeak)}$::Venus and $P_{pck-1(ΔHRE)}$::Venus, respectively. These three plasmids (100 ng/μL) were injected into wild-type animals at 100 ng/μL along with a $P_{myo-2}$::mCherry (50 ng/μL) co-injection marker.

To generate $P_{rhy-1}$::Venus transgenic animals,1510 bp upstream of the ATG of rhy-1 was amplified and cloned upstream of a venus coding sequence as above. Q5-site directed mutagenesis (Life Technologies Ltd.) was used for the removal of the ΔPeak (1170 – 617 bp upstream of start), ΔHRE1 (971 – 956 bp upstream of start), ΔHRE2 (946 – 931 bp upstream of start) and min (591 p to start) variants of the promoter. Each plasmid was injected into wild-type animals at 100 ng/μL along with a $P_{myo-2}$::mCherry (50 ng/μL) co-injection marker. Fluorescence was analyzed as above.

## Epifluorescence microscopy and image analysis

Fluorescent proteins were visualized in nematodes by mounting on 2% agarose pads with 10 mM tetramisole. All animals were synchronized by alkaline bleaching and visualized at the L4 stage. Fluorescent images of transgenic animals containing the odIs131[hif-1::gfp], $P_{pck-1}$::Venus, $P_{rhy-1}$::Venus, and/or $P_{myo-2}$::mCherry transgenes were observed using an AxioImager M1m (Carl Zeiss, Thornwood, NY). A 5X (NA 0.15), 10X (NA 0.30), or 40X (NA 1.3) PlanApo objective was used to detect fluorescence. Images were acquired with an ORCA charge-coupled device camera (Hamamatsu, Bridgewater, NJ) by using iVision software v4.1 (Biovision Technologies, Uwchlan, PA). Exposure times were chosen to capture at least 95% of the dynamic range of fluorescent intensity of all samples. Quantification was performed by obtaining outlines of nematodes using transmitted light images. The mean fluorescence intensity within each outline was calculated (after subtracting away background coverslip fluorescence using a rolling ball filter) using Fiji/ImageJ 2.1.0/1.53c[55]. The mean of raw fluorescence intensity values did not deviate more than 15% from experiment to experiment; nevertheless, a typical line was chosen for quantification, and from 20 to 100 animals (typically 50) were analyzed and pooled from two biological replicates. Averages represent Venus/mCherry ratios for animals, with individual ratio values normalized to the mean for control animals analyzed in each experiment. All data with normal distributions were analyzed with GraphPad Prism 9.3.0, in most cases using ANOVA with Dunnett's post hoc test correction for multiple comparisons.

## Quantitative RT-PCR to assess mRNA expression

Nematodes of relevant genotypes were age-synchronized by bleaching two 60 mm plates full of gravid animals. L4 animals were collected in M9 and washed twice, and then lysates were made using TRIzol® reagent (Ambion Life Technologies, Ref. 15596026) and the Direct-Zol™ RNA MiniPrep Plus (Zymo, Cat. R2070), both per manufacturer instructions. Within the Zymo protocol, residual DNA was removed with a DNAse treatment. Molecular biology grade water (Millipore, Cat. H2OMB0106) was used to elute mRNA. The nucleic acid concentration was measured using a spectrophotometer (Nanodrop or TecanPro),

and samples were diluted with water to the same concentration prior to qRT reaction preparation.

For the assessment of odIs131[hif-1::gfp] mRNA expression, two sets of primers were used to assess the expression level of the endogenous hif-1 as well as the odIs131[hif-1::gfp] transgene. The first set amplified a sequence in the 3' end of all hif-1 mRNA transcripts. The second set amplified a sequence within the hif-1(ia4) deletion. Average expression was typically normalized to untreated wild type and drawn from two or more independent, biological trials.

For assessment of HIF-1 target gene expression following hypoxia exposure, wild-type (N2) and hif-1(ia4) strains were exposed to 0 h (normoxia) and 4 h (hypoxia) of hypoxia treatment using a nitrogen-displacement hypoxia chamber (BioSpherix) pre-equilibrated to 0.5% $O_2$ for at least 2 h prior to the experiment. Four biological replicates were analyzed by qRT-PCR for each genotype and condition, with 40 ng of RNA per reaction performed with the iTaq Universal SYBR® Green One-Step Kit (BioRad, Cat.1725150) and the CFX Opus 384 Real-Time PCR System (BioRad). The results were analyzed using the ΔCq method with Actin (act-1) as a normalization control, unless otherwise specified. Samples were always normalized to an actin control run on the same plate. Per plate, reactions were performed in duplicates, and one to three independent plate technical replicates were performed. Technical replicates within the same plate that had standard deviations >0.399 between their Cq values were discarded in the analysis. Prism (Graphpad, version 9.3.0) was used for statistical testing and data representation, with specific test details found in the respective figure legends.

Primers are listed in Supplementary Data 6. They were used at 30 nM to amplify the indicated genes of interest.

## Characterizing egg laying and egg retention

All genotypes were synchronized by alkaline bleaching and arrested at L1 stage overnight in M9 buffer. Synchronized genotypes were assayed 43–46 h after reaching the L4 stage, when embryos present within adult animals were counted using a dissection microscope. From 40 to 60 animals (typically 50) were analyzed and pooled from three biological replicates. Averages represent unlaid eggs per animal.

## ChIP-seq

ChIP-Seq was performed on L4 stage nematodes for the strains OR3349 and OR3350 by the modERN/modEncode consortium as per their standard protocol[42]. Developmental synchronization was achieved by bleaching and L1 arrest. Arrested L1s were plated on NGM plates seeded with OP50 bacteria and grown at 20 °C until the L4 stage, when they were harvested by centrifugation. The pelleted nematodes were subsequently flash-frozen in liquid nitrogen and stored at −80 °C. Pellets were thawed on ice and 750 mL of FA buffer containing protease inhibitors (Roche Cat#11697498001 Complete Protease Inhibitor Cocktail Tablet, 125 μL 100 mM PMSF, and 25 μL 1 M DTT per 25 mL FA buffer) was added, and samples were then transferred to a 2 mL KONTES dounce (Kimble Chase, Vineland, NJ). Samples were dounced on ice 15 times with the small "A" pestle for two cycles with a 1 min hold between each cycle. Samples were then dounced 15 times with the large "B" pestle for four rounds with a 1 min hold between each cycle. Samples were cross-linked with 2% formaldehyde for 30 min at room temperature and then quenched with 1 M Tris pH 7.5. Samples were then sonicated to shear chromatin into 200–800 bp DNA fragments.

For each sample, 4 mg of protein lysate was immunoprecipitated using 15 μg anti-GFP antibodies (gifts of Tony Hyman and Kevin White). The polyclonal goat IgG anti-GFP antibody was validated by Western blot of immunoprecipitated material from transgenic animals expressing AMA-1::GFP[56]. Comparisons were made to control immunoprecipitations using mouse and goat IgG, as well as comparisons to input to establish background. Similarity to native profiles were determined by comparing ChIP-seq datasets between GFP-tagged

AMA-1 (Pol II) and the native protein precipitated by another antibody directly raised against native AMA-1, resulting in correlations of 0.93 and 0.95 for two biological replicates. GammaBind G Sepharose beads (GE Healthcare Life Sciences) were pre-washed and blocked in binding buffer and 0.1 mg/mL BSA. Samples were pre-cleared by adding 100 µL of 50/50 bead/buffer solution at 4 °C, followed by centrifugation. Samples of each replicate were removed and pooled to serve as total chromatin input. To each replicate, 15 µg of antibody was added to a 1:225 dilution overnight at 4 °C, followed by another overnight of incubation with the bead mix. Immunoprecipitates (IPs) were washed four times with cold lysis buffer and twice with cold TE. Pellets were resuspended in elution buffer (10 mM EDTA, 1% SDS, and 50 mM Tris-HCl pH 8) and incubated at 65 °C for 10 min. Samples were centrifuged, and supernatants were transferred to a fresh tube. Pellets were resuspended in 29% TE and 0.67% SDS and immediately centrifuged. Elution supernatants were combined and incubated at 65 °C with mild shaking overnight. Chromatin input samples were incubated at 60 °C with mild shaking overnight following the addition of Proteinase K and SDS to final concentrations of 0.1 mg/mL and 0.01%, respectively. The next day, inputs were incubated at 70 °C for 20 min. Proteinase K was added to each IP and incubated at 50 °C for 2 h. RNaseA was added to the chromatin input to a concentration of 0.017 mg/ml and incubated at 37 °C for 2 h. DNA was purified with MinElute columns (QIAGEN, Valencia, CA), eluting in 13 µL (elution buffer provided with MinElute kit). An additional 48 µL EB was added to input samples after purification. Samples were stored at −20 °C.

The enriched DNA fragments and input control (genomic DNA from the same sample) for two biological replicates were used for library preparation and sequencing. Samples were converted into libraries and multiplexed using the Ovation Ultralow DR Multiplex Systems 1–8 and 9–16 (NuGEN Technologies, San Carlos, CA) following the manufacturer's protocol, except that QIAGEN MinElute PCR purification kits were used to isolate the DNA. Briefly, 1 µL of input DNA and 10 µL of IP DNA was used to prepare sequencing libraries using NuGEN Ultralow library kits. Samples were prepared according to the manufacturer's protocol. Sequencing was performed on the Illumina HiSequation 2000/2500/4000, resulting in a range of 6.5–14.1 M single-end reads for input and both replicates for OR3349 and OR3350. Phred scores ranged from 32 to 38 out to at least 47 bases for all samples.

The Illumina sequencing data were aligned to the reference genome using the Burrows–Wheeler Aligner (BWA). Data were aligned to genome version WS245. A range of 40.4–85% of reads aligned. Peak regions significantly enriched in aligned reads were called by the SPP ChIP-seq processing pipeline standard for modERN/ModENCODE[42,57]. Peaks above an irreproducibility discovery rate (IDR) of 0.1% were used to generate final peak sets. Lowest enrichment values for OR3349 and OR3350 were 93.8 and 50.8, respectively. A broad outline of the modERN/ENCODE approach can be obtained here: https://www.encodeproject.org/pipelines/ENCPL631XPY/. Analysis tools can be obtained at GitHub: https://github.com/ENCODE-DCC/chip-seq-pipeline2/releases/tag/v1.3.5.1.

## RNA-seq

Developmentally synchronized animals were obtained by hypochlorite treatment of gravid adults and embryos hatched overnight for 15–17 h in M9. Starvation-arrested L1s were plated on NGM plates and grown at 20 °C until L4 stage. Total RNA was isolated from animals using Trizol (Invitrogen) combined with Bead Beater lysis in four biological replicates for each genotype. An mRNA library (single-end, 50-bp reads) was prepared for each sample/replicate using Illumina Truseq with PolyA selection (Genewiz or RUCDR). Libraries were sequenced across two lanes on an Illumina HiSeq2000 (GeneWiz) or an Illumina HiSeq 2500 in Rapid Run Mode (RUCDR), resulting in a range of 38–58 M single-end reads per sample, with peak mean phred scores of 39. Reads

were mapped to the *C. elegans* genome (WS245) and gene counts generated with STAR 2.5.1a. A range of 88–91% of reads aligned. Normalization and statistical analysis on gene counts were performed with EdgeR using generalized linear model functionality and tagwise dispersion estimates. Multidimensional Scaling analysis showed tight clustering within four biological replicates, with a clear separation between genotypes in which HIF-1 is active versus inactive. Likelihood ratio tests were conducted in a pairwise fashion between genotypes with a Benjamini and Hochberg correction. Transcriptomes of L4-stage animals under aerobic conditions were compared in genotypes in which HIF-1 is active, including both *egl-9(sa307)* mutants and *egl-9(sa307); hif-1(ia4); odIs131[hif-1::gfp]* mutants, as well as genotypes in which HIF-1 is inactive, including both wild type and *egl-9(sa307); hif-1(ia4)* double mutants. Genes were considered to be HIF-1-dependent if they were differentially expressed with an FDR < 0.01 in the same direction (up or down) in all four of the following pairwise comparisons: (1) *egl-9* vs. N2; (2) *egl-9* vs. *egl-9 hif-1*; (3) *odIs131; egl-9* vs. N2; (4) *odIs131; egl-9* vs. *egl-9 hif-1*. RNA-seq datasets are available at NIH/NCBI GEO through accession number GSE173581.

## Identification of direct targets using BETA basic and LOA

Two pairwise RNA-seq comparisons (wild-type vs. *egl-9* single mutants, and wild type vs. *egl-9 hif-1* double mutants rescued by the *odIs131* transgene) were analyzed, measuring differential gene expression for the same genes in both comparisons. Taking a conditional approach, the information from the first comparison (wild-type vs. *egl-9* single mutants) was examined to see if it affected interpretation in the second (wild-type vs. *egl-9 hif-1* double mutants rescued by the *odIs131* transgene). Using the approach of Luperchio et al.[40], the genes in the second comparison were split into two groups, conditional on the results in the first comparison, with one group comprising genes found to show differential expression in the first comparison, and the second group comprising genes found not to show differential expression. To estimate which genes were differentially regulated, an FDR of 0.01 was used to generate an overlapping list between the two comparisons, which was termed LOA Combo 1. An identical conditional approach was used to ask whether the information from a comparison from a separate experiment (*egl-9 hif-1* double mutants vs. *egl-9* single mutants) affected interpretation in a second comparison (*egl-9 hif-1* double mutants vs. *egl-9 hif-1* double mutants rescued with the *odIs131* transgene) from that experiment. Overlapping lists between these two comparisons were termed LOA Combo 2. BETA basic was used to identify potential direct targets for HIF-1 for the WS245 annotation of the *C. elegans* genome[41]. The following parameters were used: 15 kb from TSS, FDR cutoff of 0.01 and one-tail KS test cutoff of 0.01. The input files consisted of.bed files of IDR thresholded peaks and differential expression Log$_2$FC and FDR values for the overlap (intersection) between LOA Combo 1 and Combo 2. The HIF-1 peak/gene combinations identified this way were labeled "LOA/BETA." Analysis tools can be obtained at GitHub: https://github.com/shahlab/hypoxia-multiomics.

In a separate analysis, the overlap of the four pairwise RNA-seq comparisons was obtained without the benefit of LOA analysis (i.e., the simple intersection of the four lists of genes satisfying the FDR cutoff of 0.01 for each list). The overlapping list was used in a BETA analysis, as above, to identify potential direct targets for HIF-1. The HIF-1 peak/gene combinations identified this way were labeled "BETA only".

## Motif identification and enrichment

The sequences for the middle of each ChIP-Seq peak (+/− 100 bp, repeat masked with N) were extracted from the UCSC Genome Browser and entered into the MEME-Chip tool v5.4.1 at meme-suite.org[58]. The following parameters were used: JASPA2018 Core

Vertebrate Database (non-redundant), motif width from 6 to 15 bp, 1st order background model, STREME cutoff of 0.05.

### Identification of transcription factor occupancy at HIF-1 binding sites

Binding site coordinates (IDR Thresholded peaks, FDR < 0.01) for all transcription factors for which ChIP-seq was conducted at the L4 stage were obtained from the modENCODE repository[38]. In order to determine the number of instances these peaks overlapped with the HIF-1 ChIP-seq peaks for OR3350, we developed an R script that is available at https://github.com/shahlab/hypoxia-multiomics.

### Metabolomics

Liquid chromatography and mass spectrometry (LC-MS) were used to obtain metabolomic profiles of L4 stage wild type and *egl-9(sa307)* mutants under aerobic conditions, examining 558 metabolites in nine independent biological replicates. All genotypes were age-synchronized by alkaline bleaching, and arrested L1 larvae were plated on 100-mm plates containing standard NGM media with OP50. Animals were grown at 20 °C until L4 stage, at which point they were washed into 50 ml conical bottom tubes and allowed to settle for 15 min. Animals were then washed with 3 ×40 mL sterile M9 and centrifuged at 2000 rpm for 5 min. Five hundred μL of the final pellet was transferred to a 15-mL conical bottom tube and flash-frozen in liquid $N_2$ and stored at −80 °C until sent to Metabolon (Metabolon Ltd) for metabolomic processing. Samples were extracted with methanol, and each extract was divided into four fractions: two for analysis by two separate reverse phase (RP)/UPLC-MS/MS methods with positive ion mode electrospray ionization (ESI), one for analysis by RP/UPLC-MS/MS with negative ion mode ESI, and one for analysis by HILIC/UPLC-MS/MS with negative ion mode ESI. Raw data were extracted and peak-identified using custom software (Metabolon), then compared to a library based on authenticated standards. Statistical analysis was performed using R in ArrayStudio (Welch's *t* test, corrected for multiple comparisons).

### Paraquat survival assay

Paraquat sensitivity was performed according to the acute paraquat sensitivity assay described in Senchuk et al.[59]. Animals were placed at the L4 stage at 20 °C and survival was counted every hour until all animals were dead. From 20 to 100 animals (typically 50) per strain per trial were used for the assays, with average percent of animals surviving for 15 h drawn from five independent, biological trials.

### Hypoxia survival assays

The *pck-1* null mutants were viable under aerobic conditions, most likely because two other PEP carboxykinases, *pck-2* and *pck-3*, compensate for baseline function. Neither *pck-2* nor *pck-3* showed HIF-1-dependent regulation, suggesting *pck-1* is a specific HIF-1-induced isoform.

For hypoxia starvation, M9 solution was pre-equilibrated with 0.5% $O_2$ for at least an hour. Nematodes were grown on standard NGM plates with food. Age-synchronized L4 animals ($n \geq 20$) were then collected from plates, washed with M9 buffer, and then incubated with hypoxic M9 solution in capped tubes at 20 °C for the indicated time. FUdR (5-fluoro-2-deoxyuridine, VWR 102573-230) was added at a final concentration of 50 μM to prevent the production of progeny, which would interfere with the assay. After hypoxia starvation, nematodes were collected by centrifugate and returned to standard NGM plates under normoxia for 24 h to allow recovery. Survival of animals was counted by assessing movement and response to gentle touch by a platinum wire pick. From 20 to 100 animals (typically 50) per trial were used for the assays, with average percent of animals surviving for 48 h drawn from four independent, biological trials. A separate set of control (normoxic) animals were analyzed in the same fashion, but not exposed to hypoxia.

For embryonic survival under hypoxia, we used a previously published protocol[29]. Age-synchronized nematodes were grown on standard NGM plates with food until they reached 48 h past L4 stage, when they were lysed by hypochlorite treatment to release their in utero embryos. Embryos were placed on new NGM plates with food, counted, then exposed to a pre-equilibrated (0.5% $O_2$) hypoxia chamber for 24 h at 25 °C. Following hypoxia treatment, the plates were removed from the chamber and incubated under normoxic conditions for an additional 24 h at 25 °C. The number of unhatched embryos was counted to calculate the survival rate. From 50 to 100 embryos per trial were used for the assays, with average percent of embryos surviving drawn from six independent, biological trials. A separate set of control (normoxic) animals were analyzed in the same fashion, but not exposed to hypoxia.

### Fixation of OP50 *E. coli*

Paraformaldehyde-fixed OP50 was prepared as previously described[49]. Cultures of OP50 bacteria were grown at 37 °C with shaking in 250 mL Erlenmeyer flasks for 16 h. Freshly made paraformaldehyde stock solution was added to obtain a final concentration of 0.25%. Treated bacteria were further incubated at 37 °C with shaking for 1 h, then transferred to 50 mL conical tubes using sterile serological pipettes. Samples were centrifuged at 2500×*g* for 30 min. The supernatant was removed and 25 mL of LB was used to wash the pellet, followed by re-centrifugation for a total of four times. Pellets from washed samples were resuspended in a reduced volume of fresh LB, concentrating the media two-fold. Control (i.e., live, unfixed) cultures were prepared in an identical manner, but without the addition of paraformaldehyde. Ten microliters of the bacteria were serially diluted in a total 100 μL volume of LB 12 times. Fifty microliters from each dilution were then spread onto LB agar plates and incubated overnight at 37 °C. The number of colony-forming units (CFUs) was determined by counting the number of colonies per plate and factoring in the spread volume and dilution factor. In four independent experiments, fixation reduced the number of CFUs by at least 10 and up to 12 orders of magnitude. The bacterial cultures were used to seed NGM plates (200 μL/plate), which were allowed to dry 2 days before use.

### Metabolite and antioxidant supplementation

For the paraquat survival and hypoxia-starvation assays, nematodes were grown on standard NGM plates containing the supplement prior to collection at the L4 stage for the assay. For the hypoxia embryo survival assay, the parents of the embryos were grown on the supplement plates throughout their lifespan. The supplements included 10 mM glycolate (Sigma CDS000626), sodium pyruvate (Sigma P2256), N-acetyl-L-cysteine (Sigma A7250), or phosphoenol pyruvic (PEP) acid trisodium salt (VWR IC15187283). The same protocol was then followed as above for each assay. No supplement was included in the recovery plates.

### Lifespan analysis

*C. elegans* strains were grown at 20 °C on food for at least two generations before the experiments. For synchronization, 20–30 gravid, well-fed adults were transferred to a new NGM plate with live *E. coli* OP50-1 bacteria to lay eggs for 4–5 h, and then removed. Progeny hermaphrodite animals were maintained at 20 °C until the late L4 larval stage, and lifespan assays were counted as days after L4 stage. Animals were transferred away from their progeny to fresh OP50-seeded plates every 1–2 days until the end of their reproductive period. Survival analyses were performed using the Kaplan–Meier method, and the significance of differences between survival curves calculated using the log-rank test. Animals that showed defects due to aberrant vulval development or egg laying (e.g., bursting at the vulva, bagging, etc.) or desiccated on the side

of the dish were censored at the time of their demise. Three independent biological replicates were performed; all three gave the same mean lifespan. A representative curve with the least number of censored animals was graphed.

### Human HIF1A transcriptomics meta-analysis

Salmon-generated transcript counts from ref. 60 were filtered for studies that had at least two or more replicates. Differential gene expression analysis using DESeq2 was then performed on the resulting filtered list of 32 studies. Human genes were deemed consistent if they were upregulated with a $q$-value <0.05 in the datasets from at least 15 studies. Expression correlation and fold-change heatmaps were generated using ggpubR() and pheatmap(). The code used to generate these figures can be found at https://github.com/shahlab/hypoxia-multiomics.

### Statistical analysis

Simple calculations were done in MS Excel v16.64. Data for phenotypic analysis (e.g., survival assays, lifespan assays, egg retention assays, qRT-PCR measurements, and individual animal fluorescence measurements) were analyzed using GraphPad Prism 9.3.0. Larger datasets (e.g., RNA-seq, ChIP-seq, metabolomics) were analyzed with scripts written in R or as otherwise indicated above. Where possible, researchers were blind to genotype or experimental treatment. Statistical tests were as indicated in the figures; two-sided tests were used unless otherwise indicated. Data were tested for normality using Kolmogorov–Smirnov and adjusted for multiple comparisons as indicated in the figure legends.

### Reporting summary

Further information on research design is available in the Nature Research Reporting Summary linked to this article.

## Data availability

ChIP-seq datasets are available at NIH/NCBI GEO through accession number GSE7173333. The input file for OR3349 was GSM5266000, whereas the two replicates for OR3349 were GSM5266001 and GSM5266002. The input file for OR3350 was GSM5266003, whereas the two replicates for OR3350 were GSM5266004 and GSM5266005. BED files can be obtained and viewed using the UCSC browser at https://tinyurl.com/4pxu6vvv. Files for RNA-seq datasets are available at NIH/NCBI GEO through accession number GSE173581 [https://www.ncbi.nlm.nih.gov/geo/query/acc.cgi?acc=GSE173580]. The files GSM5271168, GSM5271169, GSM5271176, and GSM527117 contain data for four independent biological replicates for N2 wild-type nematodes. The files GSM5271170, GSM5271171, GSM5271178, and GSM5271179 contain data for four independent biological replicates for *hif-1(ia4)* mutant nematodes. The files GSM5271172, GSM5271173, GSM5271180, and GSM5271181 contain data for four independent biological replicates for *egl-9(sa307)* mutant nematodes. The files GSM5271174, GSM5271175, GSM5271182, and GSM5271183 contain data for four independent biological replicates for *egl-9(sa307) hif-1(ia4)* mutant nematodes. The files GSM5271184, GSM5271185, GSM5271186, and GSM5271187 contain data for four independent biological replicates for OR3350 nematodes. The SuperSeries file of the above subfiles is available at GSE173581. *C. elegans* ModENCODE datasets can be obtained at http://www.modencode.org/publications/worm_2010pubs/index.shtml. Source data are provided with this paper.

## Code availability

A broad outline of the modERN/ENCODE approach for identifying transcription factor ChIP-seq sites can be obtained here: https://www.encodeproject.org/pipelines/ENCPL631XPY/. Analysis tools specifically used for identifying HIF-1 ChIP-seq sites can be obtained at GitHub: https://github.com/ENCODE-DCC/chip-seq-pipeline2/releases/tag/v1.3.5.1. The code used to generate the human meta-analysis and HOT site figures can be found at https://github.com/shahlab/hypoxia-multiomics and duplicated at https://doi.org/10.5281/zenodo.7076424.

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

## Acknowledgements

We thank the TransgenOme Project for the *hif-1::gfp* fosmid, Peter Schweinsberg and Barth Grant for help with bombardment integration, Daja O'Bryant and Hazel Schubert for assistance in establishing bioinformatic pipelines, and Michelle Kudron and Valerie Reinke for their

assistance with the ChIP-seq. This work was supported by NIH grant R01GM101972 to C.R., NIH grant R35GM124976 to P.S., and a New Jersey Commission on Spinal Cord Research Postdoctoral Fellowship (CSCR13FEL001) to S.M.P.

## Author contributions

M.V., S.M.P., and C.R. designed the overall set of experiments, as well as analyzed and interpreted the data. S.M.P. and T.L.M. mediated the ChIP-seq and RNA-seq, whereas M.V., S.M.P., J.F., P.S., and C.R. analyzed the results. M.V. mediated and analyzed the metabolomics. M.V., A.P., and N.S.K. constructed and analyzed the *pck-1* promoter mutations. T.P. conducted and analyzed the qRT-PCR. M.V. and C.R. conducted hypoxia and oxidative stress survival assays, as well as wrote the manuscript.

## Competing interests

P.S. is a Scientific Advisory Board member of Trestle Biosciences, consults for Ribo-Therapeutics, and is a Director at an RNA-therapeutics startup. The remaining authors declare no competing interests.
