## [Peer Review File · Nature Communications]

The Hypoxia Response Pathway Promotes PEP Carboxykinase
And Gluconeogenesis In *C. elegans*REVIEWER COMMENTS

Reviewer #1 (Remarks to the Author):

This study employs genetic strategies in *C. elegans* to identify transcriptional targets of the HIF-1 hypoxia-inducible factor, and it identifies metabolites that are misregulated in animals that express constitutively high levels of HIF-1. The authors identify *pck-1*, a phosphoenolpyruvate carboxykinase (PEPCK) paralog, as a direct transcriptional target of HIF-1. This finding illuminates the complexity of HIF-1 functions in cellular oxygen homeostasis and energy metabolism.

1. What are the noteworthy results?

To identify transcriptional targets of HIF-1, the authors combined ChIP-seq and RNA-seq analyses. With the mutants that are available in *C. elegans*, it is possible to compare animals that lack a functional HIF-1 to those that over-express HIF-1. The authors extend understanding of HIF-1 functions by identifying direct genomic targets of HIF-1 and by comparing metabolic profiles of wild-type animals and *egl-9* mutants raised in room air conditions. This combinatorial approach shows that HIF-1 misregulation impacts many regulators of energy metabolism. The authors further investigate the importance of PCK-1, a PEPCK paralog, as a direct target of HIF-1. PEPCK is a rate-limiting enzyme in gluconeogenesis and glyceroneogenesis pathways. The authors report that *egl-9* mutants (over-expressing HIF-1, even in oxygen-rich conditions) exhibit altered levels of key metabolites, including PEP. Mutants that lack *hif-1* or *pck-1* are less able to survive hypoxia (48 hours in 0.5% oxygen), and the authors show that PEP supplementation restores hypoxia survival to wild-type levels.

2. Will the work be of significance to the field and related fields? How does it compare to the established literature?

The roles of HIF-1 in oxygen and energy homeostasis are of great interest to biomedical fields. Hypoxia and cellular responses to hypoxia are centrally important to stem cell biology, tumor development, and many other developmental or pathological processes.

As acknowledged in the manuscript, other published studies have identified genes that are mis-regulated in *hif-1* or *egl-9* backgrounds. This manuscript adds to those studies and also contributes ChIP-seq and metabolic data that illuminate the complex roles of HIF-1 in metabolic networks.

3. Does the work support the conclusions and claims, or is additional evidence needed? Are there any flaws in the data analysis, interpretation and conclusions? Is the methodology sound? Is there enough detail provided in the methods for the work to be reproduced?

The RNAseq and CHIP seq studies are sound.

It appears that the metabolic analyses reflect a single biological replicate of worms from each genetic background [egl-9 (sa307) and wild-type] grown in room air. Multiple biological replicates are needed to provide confidence that the observed differences are reproducible and significant.

It is interesting that hif-1 or pck-1 mutants benefit from growth on plates supplemented with 50mM PEP. The most parsimonious explanation is that the *C. elegans* uptake PEP directly and that this allows them to bypass a pck-1 function. Another possibility is that the bacterial food on the plates is altered by PEP. The pck-1 mutant has been shown to be sensitive to starvation (Hibshman et al eLife 2017;6:e30057). Do other metabolites also rescue pck-1 and hif-1, or is the effect specific to PEP?

Reviewer #2 (Remarks to the Author):

This manuscript reveals the direct gene targets of stabilized Hypoxia Inducible Factor (HIF-1) through the transcriptomic and RNA-seq analyses. Combining data from direct HIF-1 targets and metabolic analyses of stabilized HIF-1, they found that active HIF-1 promotes gluconeogenesis and increases oxidative and hypoxia stress resistance.

The data are clearly displayed and interpreted and support the conclusions well. The methodology is novel in that the authors compare the RNA-seq data with the CHIP-seq data to collect the direct target genes with HIF-1 binding sites; and the authors compare the RNA-seq data for the metabolism pathway enzymes and the metabolomics data for the metabolite levels. The regulated metabolites and enzymes for each metabolic pathway are well interpreted. There are several comments to the authors.

1. A primary concern is that the hypoxic response is exclusively induced by the egl-9 mutant. While egl-9 mutants do stabilize HIF-1, they aren't perfect models for hypoxic response activation from actual hypoxia. While it would be unreasonable to repeat -omics assays using hypoxia or a different hypoxia

mimetic (e.g. vhl-1 KO, which may differ due to hydroxylated prolines), validating key findings in hypoxia and/or a different mimetic would provide increased confidence that the results are robust and not egl-9/Phd mutation-dependent.

2. The authors use the CHIP-seq data and RNA-seq data to identify the direct gene targets of HIF-1. Figure 2 demonstrates that pck-1, the rate-limiting mediator of gluconeogenesis, is a direct target of HIF-1. Are other enzymes, besides pck-1, of different metabolic pathways shown in Figure 3c-j, included in the 145 direct targets of HIF-1? In other words, please clarify, based on the data, which metabolic changes are likely directly regulated by HIF-1 transcriptional activity, and which metabolic changes are indirectly regulated by downstream targets of HIF-1. There is evidence that downstream targets of HIF-1 can also regulate multiple metabolic pathways.

3. The authors proposed that pck-1 drives gluconeogenesis which increases pentose phosphate pathway flux and produces NADPH. NADPH increases reduced glutathione, which helps combat oxidative stress. NADPH also increases fatty acid synthesis. In Figure 4, the authors provide solid evidence that the oxidative or hypoxic stress resistance is hif-1 and pck-1 dependent. Have the authors tried to examine whether enzymes in the Pentose Phosphate Pathway or Glutathione Synthesis are required for oxidative or hypoxic stress resistance in the egl-9 KO background similar to egl-9(sa307); pck-1(ok2098)?

Specific comments:

1. Line 60: "some cancers", please specify.
2. Extended Data Fig. 4C: "All sites" was partially cut off by other labelling.
3. Line 128: Please give more information about PQM-1, similar to the one sentence intro of AHA-1 and SKN-1.

Reviewer #3 (Remarks to the Author):

Vora et al' article entitled "The Hypoxia Response Pathway Promotes PEP Carboxykinase and Gluconeogenesis" uses a multiomic approach to show that HIF-1 directly promotes the expression of pck-1, an enzyme that is key to a carbon storage strategy called gluconeogenesis. It also shows that this enzyme is key to survival under hypoxic conditions.

This is the most comprehensive multi-omics study that has been done for a hypoxic response (HR). And it effectively identified direct targets of HIF-1 using a highly stringent overlap analysis. However, the article falls short in two major ways. First, it does not use truly integrative statistical approaches, and therefore it falls short to show (1) all the direct targets of HIF-1 and (2) the integrated global response at the level of the transcriptome and the metabolome (3) the metabolic fluxes -predicted or measured- downstream of HIF-1 function.

All these analyses can be performed with the data they have generated, but they were -however- not performed. The second short coming of this work is that it does not represent a major step forward in our understanding of the HR or Cancer treatment. This is because gluconeogenesis has already been described as a metabolic alteration downstream of HR (in 1995) and gluconeogenesis targets are already used as anti-therapeutic targets of cancer cells that thrive under hypoxic conditions. The true novelty is to show that HIF-1 directly binds and essential enzyme of HR, pck-1. This is a relevant and important discovery, but I do not believe that merits publication in Nature Communications in its current form.

Major comments

1. In figure 1. Among their beta list authors identified 4 genes that were previously known HIF-1 targets. How many known HIF-1 targets were not identified? Is there any way to quantify the accuracy and precision of their approach?

This article uses a simple statistical method to detect shared alterations. In short: a differential accessibility and expression analysis is performed for each condition, and then a list of overlapping differential hits is created. The biggest disadvantage is that arbitrary thresholds are used to create each individual list, causing huge under-estimation of each group before overlap analysis is performed. The method is clearly highly stringent -in a good way- because it eliminates spurious binding sites such as HOT sites. However, I wonder if they are not hugely under-estimating HIF-1 genomic response. A new statistical method (Luperchio et al, eLife 2021) has been developed and used in other contexts, where a set of genes flagged as positive hits are used as prior information to test the same set of hits under a different condition. In this way, a joint analysis based on conditional p-values provide an almost 4-fold enrichment compared to the basic statistical method used in this article. I do not know how cumbersome would it be to re-do the analysis using this modern statistical method, but it would certainly improve the success rate -while still extracting information stringently-. This may improve our understanding of the functional HIF-1 binding sites in the C.elegans genome.

2. The authors -sensibly- define functional HIF-1 binding sites as those that not only show direct HIF-1 binding but that also depend on HIF-1 to be transcribed. They have done this by cleverly using a combination of mutants. I am not sure why they did not further test if the peaks identified as functional -based on this definition with HIF-1 binding sites- could also be identified under hypoxic conditions? Are these hits upregulated in the absence of O₂?

3. In Figure 2, the expression of HIF-1 targets should be also analysed under hypoxic conditions?

4. Is there a reason why simple heat maps and no joint analysis of metabolomics and transcriptomics were done? I suppose they focused their analysis on a few (145) transcriptomic hits highlighted by their BETA method. One of the benefits of performing conditional statistics to re-calculate HIF targets is that their work will gain statistical power (ie, increase the number of hits) hopefully allowing them to perform joint statistical analysis with the metabolomics data.

5. This work is important because it is the first study to combine multiple-omic frameworks in the study of HIF-1 function. As such, it should present a global analysis of all HIF datasets and not a narrow snapshot. Global transcriptomics and metabolomics datasets can be harvested using methods for integrated analysis that exploit the fact that genes and metabolites are linked through biochemical reactions and part of common pathways. Impala is one example that provide a bird's eye view through the lens of Pathway analysis; genome scale modelling that has been tailored for the analysis of post-developmental *C.elegans* (<https://doi.org/10.3389/fmolb.2019.00002>; PMID: 31953826) use metabolomics data to constrain the model, whilst using transcriptomics data to infer metabolic fluxes. This second method and it works best at visualising and predicting rate limiting steps within well annotated metabolic pathways such as the TCA cycle. What will be gained by using a more global analysis of the data is a deeper understanding of the metabolic remodelling events that happen directly or indirectly as a result of HIF function.

6. The novelty of their results is framed in the context of the "Warburg effect" which is the reliance in anaerobic glycolysis rather than oxidative phosphorylation. The Warburg effect is an important consideration for cancer treatment. However, inhibitors of pyruvate kinase are already in use as an anti-cancer treatment (example: <https://doi.org/10.3389/fonc.2013.00038>) and gluconeogenesis is a known metabolic remodelling event downstream of hypoxia (Physiol Res. 44: 257- 260, 1995). So, in essence, the novelty of this work is the discovery that HIF-1 directly bind to target genes involved in Carbon metabolism rewiring.

7. The discussion claims that "HIF-1 directly promotes the expression of the pck-1 PEP carboxykinase, a key mediator of gluconeogenesis, as well as metabolic flux through the pathway (Fig. 4d)" However, this work shows no flux analysis. Fluxes can be predicted using metabolic modelling or directly following labelled carbon. Therefore, the fluxes have only been inferred by this work.

Minor points:

1. Is there a difference among the regulatory elements? Are they all considered proximal, or are there some that can be considered distal within the 500bp interval where HRE sites are located? Are other TF preferentially bound to proximal or distal sites?

2. Overall I found legends lack enough details to be self-explanatory. For example, Figure 2 legend (B-G) Unclear in a-h if the worms are under normoxic or hypoxic conditions. This has to be clearly spelled out in the legends. (I, J) briefly indicate in the legends how binding of other TFs was identified (Modenco?)

3. Fig 2H and other bar plots throughout the article: dot plots (with a bar but showing individual values) should be used. Bar plots can be misleading.

4. Fig 2H: Blue bars show small significant change which does not make sense if the peak/binding site has been removed. How was fluorescence quantified?. Figure 2C shows some expression in the hypodermis but perhaps also in the last cell of the gut? Should worms be re-quantified without considering the tail region?

5. How did they normalise the log fold change in the heat maps in figure 3, was the normalisation done separately for transcriptomics and metabolomics?

6. Figure 3: were the transcriptomics/metabolomics datasets linked/co-harvested? This is not indicated in the methods. Because of the inherent variability of metabolomics, it is usually better for statistical analysis to link the samples. It decreases technical variability whilst preserving biological variability, which can be further harvested using network analysis.

7. Figure 4 shows that pck-1 is essential for survival under oxidative stress and starvation-hypoxic conditions. Is pck-1 essential under normoxic conditions? Can it be defined as a conditionally essential gene? Perhaps their results can be discussed in the context of recent work that has redefined gene essentiality as a gene that provides adaptive flexibility.

We thank the reviewers for their thorough reading and analysis of the manuscript, and we are pleased to hear that they all found the work to be novel, comprehensive, and important:

“this finding illuminates the complexity of HIF-1 functions in cellular oxygen homeostasis and energy metabolism.”

“data are clearly displayed and interpreted and support the conclusions well”

“methodology is novel”

“this is the most comprehensive multi-omics study that has been done for a hypoxic response”

The reviewers suggested additional experiments and revisions to make our manuscript stronger. We are quite grateful for the constructive feedback, as it has strengthened the manuscript considerably. We addressed their specific suggestions and revisions one at a time (blue italicized text below) with our responses (black, non-italicized text).

Reviewer #1 (Remarks to the Author):

*This study employs genetic strategies in *C. elegans* to identify transcriptional targets of the HIF-1 hypoxia-inducible factor, and it identifies metabolites that are misregulated in animals that express constitutively high levels of HIF-1. The authors identify *pck-1*, a phosphoenolpyruvate carboxykinase (PEPCK) paralog, as a direct transcriptional target of HIF-1. This finding illuminates the complexity of HIF-1 functions in cellular oxygen homeostasis and energy metabolism.*

*1. What are the noteworthy results? To identify transcriptional targets of HIF-1, the authors combined ChIP-seq and RNA-seq analyses. With the mutants that are available in *C. elegans*, it is possible to compare animals that lack a functional HIF-1 to those that over-express HIF-1. The authors extend understanding of HIF-1 functions by identifying direct genomic targets of HIF-1 and by comparing metabolic profiles of wild-type animals and *egl-9* mutants raised in room air conditions. This combinatorial approach shows that HIF-1 misregulation impacts many regulators of energy metabolism. The authors further investigate the importance of PCK-1, a PEPCK paralog, as a direct target of HIF-1. PEPCK is a rate-limiting enzyme in gluconeogenesis and glyceroneogenesis pathways. The authors report that *egl-9* mutants (over-expressing HIF-1, even in oxygen-rich conditions) exhibit altered levels of key metabolites, including PEP. Mutants that lack *hif-1* or *pck-1* are less able to survive hypoxia (48 hours in 0.5% oxygen), and the authors show that PEP supplementation restores hypoxia survival to wild-type levels.*

*2. Will the work be of significance to the field and related fields? How does it compare to the established literature? The roles of HIF-1 in oxygen and energy homeostasis are of great interest to biomedical fields. Hypoxia and cellular responses to hypoxia are centrally important to stem cell biology, tumor development, and many other developmental or pathological processes. As acknowledged in the manuscript, other published studies have identified genes that are mis-regulated in *hif-1* or *egl-9* backgrounds. This manuscript adds to those studies and also contributes ChIP-seq and metabolic data that illuminate the complex roles of HIF-1 in metabolic networks.*

AUTHORS' RESPONSE: We thank this reviewer for a thorough reading of our manuscript and for their enthusiasm as to its significance.

3. Does the work support the conclusions and claims, or is additional evidence needed? Are there any flaws in the data analysis, interpretation and conclusions? Is the methodology sound? Is there enough detail provided in the methods for the work to be reproduced?

The RNAseq and ChIP seq studies are sound.

It appears that the metabolic analyses reflect a single biological replicate of worms from each genetic background [egl-9 (sa307) and wild-type] grown in room air. Multiple biological replicates are needed to provide confidence that the observed differences are reproducible and significant.

AUTHORS' RESPONSE: We agree with the reviewer that multiple biological replicates increase the confidence of the result; indeed, the metabolomics data we presented represents 9 independent biological replicates. We apologize for not making this clear in the initial version of the manuscript, and we have since clarified this in the Materials and Methods of the revision.

It is interesting that hif-1 or pck-1 mutants benefit from growth on plates supplemented with 50mM PEP. The most parsimonious explanation is that the C. elegans uptake PEP directly and that this allows them to bypass a pck-1 function. Another possibility is that the bacterial food on the plates is altered by PEP. The pck-1 mutant has been shown to be sensitive to starvation (Hibshman et al eLife 2017;6:e30057). Do other metabolites also rescue pck-1 and hif-1, or is the effect specific to PEP?

AUTHORS' RESPONSE: The alternative explanation provided by the reviewer is valid, and we addressed this issue in two ways. First, we adopted a feeding protocol from Beydoun et al. (PMID 33637830), which used gentle paraformaldehyde fixation to kill and metabolically neutralize the OP50 *E. coli* food source for *C. elegans* without dramatically affecting nematode lifespan and growth. We then repeated our metabolic supplementation experiments, including with supplemental PEP feeding, using this metabolically inert food source. We found the same results regardless of whether live or fixed *E. coli* food was used. This method allowed us to ensure that the bacterial food was not a secondary vector through which the supplemental metabolites acted, thus allowing us to remove this variable from the obtained results and strengthen our conclusions. These experiments have been added to the manuscript (Figure 4 and Extended Data Figure 6).

Second, we examined additional metabolites to directly test the hypothesis that HIF-1, through its upregulation of PCK-1, promotes hypoxia survival by promoting gluconeogenesis and an antioxidant response. We also added an additional hypoxia survival assay to our analysis. We found that supplemental antioxidants and PEP both robustly rescue *hif-1* and *pck-1* mutants, whereas supplemental pyruvate, which is a metabolite that falls upstream of PCK-1 in the gluconeogenesis pathway, does not rescue *hif-1* or *pck-1* mutants. These experiments have been added to the manuscript (Figure 4 and Extended Data Figure 6).

Reviewer #2 (Remarks to the Author):

This manuscript reveals the direct gene targets of stabilized Hypoxia Inducible Factor (HIF-1) through the transcriptomic and RNA-seq analyses. Combining data from direct HIF-1 targets and metabolic analyses of stabilized HIF-1, they found that active HIF-1 promotes gluconeogenesis and increases oxidative and hypoxia stress resistance.

The data are clearly displayed and interpreted and support the conclusions well. The methodology is novel in that the authors compare the RNA-seq data with the ChIP-seq data to collect the direct target genes with HIF-1 binding sites; and the authors compare the RNA-seq data for the metabolism pathway enzymes and the metabolomics data for the metabolite levels. The regulated metabolites and enzymes for each metabolic pathway are well interpreted. There are several comments to the authors.

1. A primary concern is that the hypoxic response is exclusively induced by the egl-9 mutant. While egl-9 mutants do stabilize HIF-1, they aren't perfect models for hypoxic response activation from actual hypoxia. While it would be unreasonable to repeat -omics assays using hypoxia or a different hypoxia mimetic (e.g. vhl-1 KO, which may differ due to hydroxylated prolines), validating key findings in hypoxia and/or a different mimetic would provide increased confidence that the results are robust and not egl-9/Phd mutation-dependent.

AUTHORS' RESPONSE: The reviewer makes an important point: activation of HIF-1 under aerobic conditions versus hypoxic conditions, while likely to have significant overlapping effects, are not guaranteed to be identical. We agree. We conducted additional experiments in which either wild type or *hif-1* mutant nematodes were exposed to hypoxia and collected for RNA isolation to see if some of the key genes regulated in *egl-9* mutants also showed differential regulation in response to hypoxia. We used qRT-PCR to measure the mRNA levels of multiple direct target genes, including *pck-1*, that were identified in our analysis of the *egl-9* mutants. All the genes examined showed the same HIF-1-dependent differential regulation in response to hypoxia that they showed in *egl-9* mutants cultured under aerobic conditions. This data has been added to the paper as Figure 4. Doubtless, there will be genes showing differential regulation unique to either *egl-9* mutants or hypoxia treatment; however, identifying those genes will require a more extensive Omics analysis, which (as the reviewer points out) would be unreasonable to request for a revision in this manuscript.

2. The authors use the ChIP-seq data and RNA-seq data to identify the direct gene targets of HIF-1. Figure 2 demonstrates that pck-1, the rate-limiting mediator of gluconeogenesis, is a direct target of HIF-1. Are other enzymes, besides pck-1, of different metabolic pathways shown in Figure 3c-j, included in the 145 direct targets of HIF-1? In other words, please clarify, based on the data, which metabolic changes are likely directly regulated by HIF-1 transcriptional activity, and which metabolic changes are indirectly regulated by downstream targets of HIF-1. There is evidence that downstream targets of HIF-1 can also regulate multiple metabolic pathways.

AUTHORS' RESPONSE: Yes, other enzymes in the pathways displayed in Figure 3 are direct targets of HIF-1. We indicated these with yellow arrowheads. Our goal with Figure 3 was to provide a "bird's eye" view of three important elements of each metabolic pathway: (1) HIF-1-induced changes in gene expression for enzymes in each pathway, (2) HIF-1-induced changes in metabolite levels catalyzed by those pathway enzymes, and (3) identification of which of those pathway enzymes are direct HIF-1 targets. The first two elements are indicated by heat map color scoring. The last element (i.e., HIF-1 direct targets) are indicated by yellow arrowheads. We focused on *pck-1* specifically for the remainder of the paper because of its specific, pivotal, and rate-limiting role in gluconeogenesis.

3. The authors proposed that pck-1 drives gluconeogenesis which increases pentose phosphate pathway flux and produces NADPH. NADPH increases reduced glutathione, which helps combat oxidative stress. NADPH also increases fatty acid synthesis. In Figure 4, the authors provide solid evidence that the oxidative or hypoxic stress resistance is hif-1 and pck-1 dependent. Have the authors tried to examine whether enzymes in the Pentose Phosphate Pathway or Glutathione Synthesis are required for oxidative or hypoxic stress resistance in the egl-9 KO background similar to egl-9(sa307); pck-1(ok2098)?

AUTHORS' RESPONSE: This is a challenging issue to address, as mutants for the PPP and glutathione pathways tend to be lethal for reasons independent of their role in the hypoxia response. In many cases, the enzymes involved in the antioxidant response comprise large families of paralogs, resulting in functional redundancy, making phenotypic interpretation of mutants difficult. Instead, we approached this question by testing whether antioxidant supplementation, by itself, was sufficient to rescue *hif-1* or *pck-1* mutants from hypoxic stress. The idea here is that if *hif-1* or *pck-1* mutants are defective in mounting the full ensemble of enzymes that mediate antioxidant response during hypoxia, then we should be able to relieve these mutants of their oxidative stress by providing them with exogenous antioxidants. We found this to be the case, and we have added these experiments to the manuscript.

Specific comments:

1. Line 60: "some cancers", please specify.

AUTHORS' RESPONSE: The link between hypoxia, HIF1, and cancer is well established (a PubMed search for articles using "hypoxia" and "cancer" as query terms identifies 34,000 publications, 6,255 of which are review articles). Mutations leading to HIF1-mediated changes in glucose utilization have been observed in clear cell renal carcinoma, paraganglioma, and pheochromocytoma, leukemia, and lung cancer. Hypoxia as a hallmark of the tumor microenvironment was first described in lung carcinoma by Thomlinson and Gray in the 1950s, then demonstrated in solid tumors from breast, pancreas, brain, liver, stomach, cervix, ovary, head-and-neck, prostate, bladder, kidney, skin, and colon. To clarify this point, we have replaced "some cancers" with "cancer." For the sake of minimizing total word count, we have not added the list of specific cancers, instead citing review articles that expand on this issue.

2. Extended Data Fig. 4C: "All sites" was partially cut off by other labelling.

AUTHORS' RESPONSE: We thank this reviewer for catching this error, which we have corrected.

3. Line 128: Please give more information about PQM-1, similar to the one sentence intro of AHA-1 and SKN-1.

AUTHORS' RESPONSE: We have replaced the previous description with one that provides rationale for why we examined PQM-1: "By contrast, sites bound by PQM-1, a zinc finger transcriptional antagonist of DAF-16/FOXO, were underrepresented near HIF-1 sites, consistent with the increased hypoxia survival observed in *pqm-1* mutants."

Reviewer #3 (Remarks to the Author):

Vora et al' article entitled "The Hypoxia Response Pathway Promotes PEP Carboxykinase and Gluconeogenesis" uses a multiomic approach to show that HIF-1 directly promotes the expression of pck-1, an enzyme that is key to a carbon storage strategy called gluconeogenesis. It also shows that this enzyme is key to survival under hypoxic conditions.

This is the most comprehensive multi-omics study that has been done for a hypoxic response (HR). And it effectively identified direct targets of HIF-1 using a highly stringent overlap analysis. However, the article falls short in two major ways. First, it does not use truly integrative statistical approaches, and therefore it falls short to show (1) all the direct targets of HIF-1 and (2) the integrated global response at the level of the transcriptome and the metabolome (3) the metabolic fluxes -predicted or measured- downstream of HIF-1 function. All these analyses can be performed with the data they have generated, but they were - however- not performed. The second short coming of this work is that it does not represent a major step forward in our understanding of the HR or Cancer treatment. This is because gluconeogenesis has already been described as a metabolic alteration downstream of HR (in 1995) and gluconeogenesis targets are already used as anti-therapeutic targets of cancer cells that thrive under hypoxic conditions. The true novelty is to show that HIF-1 directly binds and essential enzyme of HR, pck-1. This is a relevant and important discovery, but I do not believe that merits publication in Nature Communications in its current form.

AUTHORS' RESPONSE: We thank the reviewer for recognizing the comprehensiveness of our analysis. More importantly, we thank the reviewer for suggesting the analysis method of Luperchio et al., which was published while our manuscript was under its initial review. As described below, we used the integrative statistical methodology of Luperchio et al. to identify additional HIF-1 targets. Although the nature of those additional targets has not changed our overview of how HIF-1 activation reprograms metabolism, it definitely increased the comprehensiveness of the study and provided an even better and more physiologically specific methodology for identifying genes directly regulated by nearby transcription factor binding sites identified by ChIP-seq. We expect that many future publications will use this approach to analyze ChIP-seq and other DNA-binding-protein Omics data, citing our manuscript as well as that of Luperchio et al.

It is true that there are previous publications that make the connection between hypoxia and gluconeogenesis; however, this connection has been understudied (e.g., only 51 publications in PubMed using the terms "hypoxia" and "gluconeogenesis") compared to connections between hypoxia and other physiological processes like glycolysis (836 publications), erythropoiesis (479 publications), and angiogenesis (2,366 publications). Some researchers have made the connection between gluconeogenesis, movement of carbon through the PPP, and redox homeostasis, but the focus of these studies have been on glycogenolysis as the driving force rather than upstream gluconeogenesis regulation by HIF1. The strength of our manuscript is that it goes beyond what is known in showing that HIF-1 directly binds to and regulates the expression of PCK (as this reviewer points out), a key mediator of gluconeogenesis, to show how redox homeostasis is a physiological outcome of that regulatory interaction important for survival of hypoxic stress. Whereas this novel finding

has implications for cancer treatment, as suggested by the reviewer, it also has implications for other diseases involving hypoxia (e.g., heart disease, stroke, COPD, cerebral palsy, pulmonary hypertension, COVID-19, et cetera) as well as for our understanding of the evolution of aerobic organisms and their ability to adapt to terrestrial and aquatic habitats with variable oxygen availability. Unequivocal identification of HIF-1 direct and indirect targets, careful dissection of the downstream metabolomic implications of the hypoxia response (only a part of which was examined in the study of PCK-1), and broad-reaching science and clinical impact is exactly why we think our work is appropriate for publication in *Nature Communications*.

Major comments

1. In figure 1. Among their beta list authors identified 4 genes that were previously known HIF-1 targets. How many known HIF-1 targets were not identified? Is there any way to quantify the accuracy and precision of their approach?

AUTHORS' RESPONSE: No direct targets have been identified in *C. elegans*. The four best studied genes known to show differential regulation in the face of HIF-1 activation were not known to be direct targets in any organism prior to our work. There have been several studies analyzing differential gene expression under conditions when HIF-1 is activated; however, these were not paired with studies identifying HIF-1 binding sites. Moreover, they were conducted at either different temperatures or different stages of development compared to conditions for our study, so there are biological reasons why the sets of genes identified in these studies might not perfectly overlap with our set of identified genes. Nevertheless, there are ten genes that consistently appear in all differential expression datasets available (using strict criteria of $q < 0.01$, with 2-fold more upregulation by HIF-1): *rhy-1*, *nhr-57*, *tyr-2*, *oac-54*, F22B5.4, *phy-2*, M05D6.5, *cyp-36A1*, *egl-9*, and *fmo-4*. All but one of these (*tyr-2*) are found as DEGs upregulated by HIF-1 in our analysis.

One way we have tried to validate our work is that we conducted additional experiments in which either wild type or *hif-1* mutant nematodes were exposed to hypoxia and collected for RNA isolation to see if some of the key genes regulated in *egl-9* mutants also showed differential regulation in response to hypoxia. As described in our response to one of the other reviewers, we used qRT-PCR to measure the mRNA levels of multiple direct target genes, including *pck-1*, that were identified in our analysis of the *egl-9* mutants. All genes examined showed the same HIF-1-dependent differential regulation in response to hypoxia that they showed in *egl-9* mutants cultured under aerobic conditions. This data was added to the manuscript (Figure 4 and Extended Data Figure 5).

This article uses a simple statistical method to detect shared alterations. In short: a differential accessibility and expression analysis is performed for each condition, and then a list of overlapping differential hits is created. The biggest disadvantage is that arbitrary thresholds are used to create each individual list, causing huge under-estimation of each group before overlap analysis is performed. The method is clearly highly stringent -in a good way- because it eliminates spurious binding sites such as HOT sites. However, I wonder if they are not hugely under-estimating HIF-1 genomic response. A new statistical method (Luperchio et al, eLife 2021)

has been developed and used in other contexts, where a set of genes flagged as positive hits are used as prior information to test the same set of hits under a different condition. In this way, a joint analysis based on conditional p-values provide an almost 4-fold enrichment compared to the basic statistical method used in this article. I do not know how cumbersome would it be to re-do the analysis using this modern statistical method, but it would certainly improve the success rate -while still extracting information stringently-. This may improve our understanding of the functional HIF-1 binding sites in the C.elegans genome.

AUTHORS' RESPONSE: We thank the reviewer for this outstanding suggestion – an excellent example of how the review process, when applied constructively, can augment a paper and lead to new results. As mentioned previously, Luperchio et al. was published while our manuscript was under review, so we did not have the benefit of having read that paper. However, we have gone back and integrated the approach outlined in Luperchio et al. with our analysis using BETA to leverage RNA-seq expression data for identifying genes directly regulated by nearby ChIP-seq binding sites. The integration of this methodology allowed us to identify additional direct targets of HIF-1. Moreover, it sharpened the specificity of our approach, as indicated by a significant drop in the number of spurious binding sites such as HOT sites in our analysis. We hope to push the model organism community, including the ModENCODE/modERN project, to adopt these innovative approaches going forward, and we think our publication, combined with Luperchio et al., will provide a major foundation towards that effort.

2. The authors -sensibly- define functional HIF-1 binding sites as those that not only show direct HIF-1 binding but that also depend on HIF-1 to be transcribed. They have done this by cleverly using a combination of mutants. I am not sure why they did not further test if the peaks identified as functional -based on this definition with HIF-1 binding sites- could also be identified under hypoxic conditions? Are these hits upregulated in the absence of O₂?

AUTHORS' RESPONSE: We have chosen a handful of HIF-1 target genes identified in our RNA-seq/ChIP-seq analysis of *egl-9* mutants and used qRT-PCR to measure changes in their expression in response to hypoxia (Extended Data Figure 5). All these genes showed HIF-1-dependent changes in expression in response to hypoxia. We have added this data to the manuscript. There are certainly hundreds of additional HIF-1 binding peaks that remain to be tested, but full validation would be beyond the scope of this manuscript.

3. In Figure 2, the expression of HIF-1 targets should be also analyzed under hypoxic conditions?

AUTHORS' RESPONSE: We interrogated the expression of endogenous *pck-1* directly using qRT-PCR and found it to also show HIF-1-dependent upregulation in response to hypoxia. We have added this data to the manuscript (Extended Data Figure 5).

4. Is there a reason why simple heat maps and no joint analysis of metabolomics and transcriptomics were done? I suppose they focused their analysis on a few (145) transcriptomic

hits highlighted by their BETA method. One of the benefits of performing conditional statistics to re-calculate HIF targets is that their work will gain statistical power (ie, increase the number of hits) hopefully allowing them to perform joint statistical analysis with the metabolomics data.

AUTHORS' RESPONSE: The Luperchio et al. analysis identified several additional enzymes involved in gluconeogenesis and the citric acid cycle, which we have added to our integrated metabolomics figure (Figure 3) as well as for all other analyses performed in the paper.

5. This work is important because it is the first study to combine multiple-omic frameworks in the study of HIF-1 function. As such, it should present a global analysis of all HIF datasets and not a narrow snap-shot. Global transcriptomics and metabolomics datasets can be harvested using methods for integrated analysis that exploit the fact that genes and metabolites are linked through biochemical reactions and part of common pathways. Impala is one example that provide a bird's eye view through the lens of Pathway analysis; genome scale modelling that has been tailored for the analysis of post-developmental C.elegans (<https://doi.org/10.3389/fmolb.2019.00002>; PMID: 31953826) use metabolomics data to constrain the model, whilst using transcriptomics data to infer metabolic fluxes. This second method and it works best at visualising and predicting rate limiting steps within well annotated metabolic pathways such as the TCA cycle. What will be gained by using a more global analysis of the data is a deeper understanding of the metabolic remodelling events that happen directly or indirectly as a result of HIF function.

AUTHORS' RESPONSE: Flux Based Analysis (FBA) tends to do a poor job of recapitulating the movement of metabolites in intact animals. The reviewer highlights Hastings et al. as an example for how to integrate transcriptomics data with whole genome models for the computational prediction of metabolic fluxes. Yet Hastings et al. highlight that there are two assumptions that are critical to FBA: the system has reached steady state and the system has been optimized through evolution to achieve biomass production. We do not believe that we can make either assumption for animals in which a stress response pathway like HIF-1 is activated. As Hastings et al. state, "For FBA to be useful, it must provide mechanistic insights into metabolism and the accuracy of the predictions can be benchmarked against many features including gene essentiality (Opdam et al., 2017) and recapitulation of metabolic states when a certain reaction has been removed (Gebauer et al., 2016)." To that point, we feel that our functional analysis of *pck-1* mutants, including their rescue for hypoxia survival by supplementation with the downstream metabolite PEP and antioxidants, is more informative than FBA modeling.

One way to increase the accuracy of FBA models would be to add a time component to such a study in the future, as this would allow longitudinal metabolic analysis in response to hypoxia onset, monitoring changes in gene expression and metabolite levels as time passes during hypoxia exposure. Paired with the appropriate metabolic mutants, this could provide a more detailed description of metabolic flux and how metabolism changes concordant with changes in gene expression. However, such a study would be far beyond the scope of our current manuscript.

6. *The novelty of their results is framed in the context of the “Warburg effect” which is the reliance in anaerobic glycolysis rather than oxidative phosphorylation. The Warburg effect is an important consideration for cancer treatment. However, inhibitors of pyruvate kinase are already in use as an anti-cancer treatment (example: <https://doi.org/10.3389/fonc.2013.00038>) and gluconeogenesis is a known metabolic remodelling event downstream of hypoxia (Physiol Res. 44: 257- 260, 1995). So, in essence, the novelty of this work is the discovery that HIF-1 directly bind to target genes involved in Carbon metabolism rewiring.*

AUTHORS’ RESPONSE: It is true that inhibitors of Pyruvate Dehydrogenase Kinase (PDK), which is often activated in cancers to reduce flux from glycolysis into mitochondrial respiration by inhibiting pyruvate dehydrogenase, are used in cancer treatment. However, the therapeutic strategy behind using these inhibitors is the reactivation of mitochondrial metabolism in these tumors, resulting in reactive oxygen species generation, the consequential deactivation of proliferative factors, and the activation of pro-apoptotic factors.

Our study is not framed around validating that HIF-1 promotes the *Warburg effect*. Rather, our work highlights the role of *gluconeogenesis and redox homeostasis* as a metabolic mechanism acting downstream of HIF-1 activation. There are a few papers showing a link between hypoxia and gluconeogenesis, but the physiological function of increased gluconeogenesis in response to hypoxia is not well understood. Our work is important because it extends our understanding of the physiological function of this link and provides a rationale for this flux: redox homeostasis. The ability of supplemental antioxidants to rescue hypoxia survival of nematodes lacking either HIF-1 or PCK-1 strongly demonstrates this point, which has broader consequences beyond cancer biology.

7. *The discussion claims that “HIF-1 directly promotes the expression of the pck-1 PEP carboxykinase, a key mediator of gluconeogenesis, as well as metabolic flux through the pathway (Fig. 4d)” However, this work shows no flux analysis. Fluxes can be predicted using metabolic modelling or directly following labelled carbon. Therefore, the fluxes have only been inferred by this work.*

AUTHORS’ RESPONSE: Metabolic modeling has limitations, as discussed above. A better alternative would be metabolic flux analysis using labelled metabolites; however, this is best performed in cultured cells that more easily and quickly take up such metabolites. Metabolite uptake by live nematodes does not occur on the timescale required to perform true metabolic flux experiments. To this end, we have added a statement in the discussion section of the manuscript highlighting this limitation: “True metabolic flux analysis requires experiments that follow labeled metabolites, which cannot be done in *C. elegans*.”

Minor points:

1. *Is there a difference among the regulatory elements? Are they all considered proximal, or are there some that can be considered distal within the 500bp interval where HRE sites are located? Are other TF preferentially bound to proximal or distal sites?*

AUTHORS' RESPONSE: Whether different transcription factor binding sites in *C. elegans* act distally versus proximally is poorly understood. The compact genome of the organism makes this a difficult question to study. ModENCODE/modERN studies suggest that most transcription factors and their binding sites in *C. elegans* are quite close to the start of transcription (TSS), typically within 500 bps or closer, and either upstream or downstream of the TSS. However, the genes inferred to be regulated by these binding sites were identified by a nearest-neighbor approach rather than through the use of functional data like RNA-seq analysis obtained under conditions that induce the activity of that given transcription factor. Thus, one should not be surprised when transcription factor binding sites cluster near the TSS of presumed target genes when TSS proximity is used as the criteria for identifying those target genes as such. Indeed, HIF-1 binding site proximity to the TSS showed a similar pattern to that observed for other transcription factors analyzed by ModENCODE/modERN using the nearest neighbor approach to call targets. However, when we used Luperchio analysis coupled with BETA analysis to identify functional direct targets rather than simply nearest neighbor, the distance of those sites from the TSS of their target gene expanded beyond 500 bps, consistent with many of those factors likely acting as distal regulators. We have updated Extended Figure 2 as well as the text to include this new data and analysis in the manuscript.

2. Overall I found legends lack enough details to be self-explanatory. For example, Figure 2 legend (B-G) Unclear in a-h if the worms are under normoxic or hypoxic conditions. This has to be clearly spelled out in the legends. (I, J) briefly indicate in the legends how binding of other TFs was identified (Modenco?)

AUTHORS' RESPONSE: We apologize for the lack of clarity in the figure legends. Data from Figure 2b-g were obtained under normoxic conditions. Data from Figure 2j came from the modENCODE/modERN consortium. We have made these changes in the figure. Hypoxia versus normoxia has been more clearly noted in the remaining figures.

3. Fig 2H and other bar plots throughout the article: dot plots (with a bar but showing individual values) should be used. Bar plots can be misleading.

AUTHORS' RESPONSE: We have incorporated dot plots where appropriate in the principal figures.

4. Fig 2H: Blue bars show small significant change which does not make sense if the peak/binding site has been removed. How was fluorescence quantified?. Figure2C shows some expression in the hypodermis but perhaps also in the last cell of the gut? Should worms be re-quantified without considering the tail region?

AUTHORS' RESPONSE: Fluorescence quantification is described in the Materials and Methods section. We note that a *statistically significant* difference does not mean that the *magnitude* of that difference is biologically meaningful. The average decrease in $P_{pck-1(\Delta Peak)}::Venus$ expression in *egl-9* mutants relative to wild type is only about 10%, which is unlikely to be as

biologically meaningful as the almost four-fold drop in $P_{pck-1(+)}::Venus$ expression in *egl-9 hif-1* double mutants compared to *egl-9* single mutants.

5. How did they normalise the log fold change in the heat maps in figure 3, was the normalisation done separately for transcriptomics and metabolomics?

AUTHORS' RESPONSE: Yes, the log fold changes were done separately for the transcriptomics and metabolomics.

6. Figure 3: were the transcriptomics/metabolomics datasets linked/co-harvested? This is not indicated in the methods. Because of the inherent variability of metabolomics, it is usually better for statistical analysis to link the samples. It decreases technical variability whilst preserving biological variability, which can be further harvested using network analysis.

AUTHORS' RESPONSE: Ideally, we would have liked to have co-harvested for our transcriptomics and metabolomics analysis for the reasons the reviewer suggested. However, we did not do this for reasons of cost. The effect sizes for differences in metabolite levels observed in published *C. elegans* metabolomics studies have been more subtle than effect sizes for differences in transcript levels observed in published *C. elegans* transcriptomics studies. Thus, we reasoned that the metabolomic studies would require a larger number of biological replicates compared to what would be required for transcriptomics. To match that larger number of biological replicates performed for the metabolomics (9 replicates) for the transcriptomics analysis (4 replicates) would have been cost-prohibitive.

7. Figure 4 shows that pck-1 is essential for survival under oxidative stress and starvation-hypoxic conditions. Is pck-1 essential under normoxic conditions? Can it be defined as a conditionally essential gene? Perhaps their results can be discussed in the context of recent work that has redefined gene essentiality as a gene that provides adaptive flexibility.

AUTHORS' RESPONSE: PCK-1 is not essential (in the classic sense of the word) under normoxic conditions in a laboratory setting. We agree that it could be defined as conditionally essential, particularly for *C. elegans* growth in its native environment of the soil. Although this is an interesting topic, we prefer not to add discussion about this issue to the text because (1) word limits for a *Nature Communication* article are strict, and (2) it would detract from the main theme of the manuscript.

REVIEWERS' COMMENTS

Reviewer #1 (Remarks to the Author):

The authors have addressed reviewer concerns by conducting and including additional controls and analyses. The methodology is sound, and this revised manuscript is a valuable and significant addition to the literature.

Reviewer #2 (Remarks to the Author):

The authors have addressed all my questions well. This manuscript will be a valuable addition to the field.

Reviewer #3 (Remarks to the Author):

I believe that the authors performed many of the analysis and the experiments requested by reviewers and as a result, the manuscript is better suited for publication in its current form. There are two areas that could still be improved. The first is that the significance of their findings is better argued in the rebuttal letters than in the article itself. I would like to see their writing nailing down the significance of their findings improved. The second is that it would be very useful if they add some quantitative analysis in the increased specificity of their integrative ChIP-seq/RNA-seq analysis by adding for example a quantitation of the hot sites. The more quantitative data they can flush out of their approach, the more convincing their results will be and more researchers will cite this work for this reason. Finally, it is a pity that all graphs are depicted as bar plots, these should be banned from high profile papers, researchers doing high flying research should set the standard high and use dots per replicate (and an uncolored bar to guide the reader)

We thank the reviewers for their thorough reading and analysis of the manuscript, and we are pleased to hear that they all found the work to be novel, comprehensive, and important.

The reviewers unanimously agreed that the additional analysis and experiments we performed made the manuscript suitable for publication. We are quite grateful for the constructive feedback, as it has strengthened the manuscript considerably. We addressed their specific suggestions and revisions one at a time (blue italicized text below) with our responses (black, non-italicized text).

Reviewer #1 (Remarks to the Author):

The authors have addressed reviewer concerns by conducting and including additional controls and analyses. The methodology is sound, and this revised manuscript is a valuable and significant addition to the literature.

AUTHORS' RESPONSE: We thank this reviewer for a thorough reading of our manuscript and for their enthusiasm as to its significance.

Reviewer #2 (Remarks to the Author):

The authors have addressed all my questions well. This manuscript will be a valuable addition to the field.

AUTHORS' RESPONSE: We thank this reviewer for a thorough reading of our manuscript and for their enthusiasm as to its value to the field.

Reviewer #3 (Remarks to the Author):

I believe that the authors performed many of the analysis and the experiments requested by reviewers and as a result, the manuscript is better suited for publication in its current form.

AUTHORS' RESPONSE: We thank this reviewer for a thorough reading of our manuscript and for their enthusiasm as to its suitability for publication.

There are two areas that could still be improved. The first is that the significance of their findings is better argued in the rebuttal letters than in the article itself. I would like to see their writing nailing down the significance of their findings improved.

AUTHORS' RESPONSE: We have moved key elements of the rebuttal letter into the discussion section of the main manuscript to address the reviewer's concern.

The second is that it would be very useful if they add some quantitative analysis in the increased specificity of their integrative ChIP-seq/RNA-seq analysis by adding for example a quantitation of the hot sites. The more quantitative data they can flush out of their approach, the more convincing their results will be and more researchers will cite this work for this reason.

AUTHORS' RESPONSE: We have added a statistical analysis to the HOT site data described in Figure 2. This is the new Supplemental Figure 5.

Finally, it is a pity that all graphs are depicted as bar plots, these should be banned from high profile papers, researchers doing high flying research should set the standard high and use dots per replicate (and an uncolored bar to guide the reader).

AUTHORS' RESPONSE: We were a bit confused regarding this concern, as we changed all the graphs in the primary figures to dot plots (with dots indicating replicates) in the previous revisions. Perhaps the reviewer is referring to the graphs in the supplemental figures? We have now converted those to dot plots, too.